# Sleep-like unsupervised replay reduces catastrophic forgetting in artificial neural networks

Timothy Tadros [1,2], Giri P. Krishnan[2], Ramyaa Ramyaa[3] & Maxim Bazhenov [1,2] ✉

Artificial neural networks are known to suffer from catastrophic forgetting: when learning multiple tasks sequentially, they perform well on the most recent task at the expense of previously learned tasks. In the brain, sleep is known to play an important role in incremental learning by replaying recent and old conflicting memory traces. Here we tested the hypothesis that implementing a sleep-like phase in artificial neural networks can protect old memories during new training and alleviate catastrophic forgetting. Sleep was implemented as off-line training with local unsupervised Hebbian plasticity rules and noisy input. In an incremental learning framework, sleep was able to recover old tasks that were otherwise forgotten. Previously learned memories were replayed spontaneously during sleep, forming unique representations for each class of inputs. Representational sparseness and neuronal activity corresponding to the old tasks increased while new task related activity decreased. The study suggests that spontaneous replay simulating sleep-like dynamics can alleviate catastrophic forgetting in artificial neural networks.

Humans and animals have a remarkable ability to learn continuously and to incorporate new data into their corpus of existing knowledge. In contrast, artificial neural networks (ANNs) suffer from catastrophic forgetting whereby they achieve optimal performance on newer tasks at the expense of performance on previously learned tasks[1–4]. This dichotomy between continual learning in biological brains and catastrophic forgetting in machine learning models has given rise to the stability-plasticity dilemma[5–8]. On the one hand, a network must be plastic such that the parameters of the network can change in order to accurately represent and respond to new tasks. On the other hand, a network must be stable such that it maintains knowledge of older tasks. Although deep neural networks[9] can achieve supra-human level of performance on tasks ranging from complex games[10] to image recognition[11], they lie at a suboptimal point on the stability-plasticity spectrum.

Sleep has been hypothesized to play an important role in memory consolidation and generalization of knowledge in biological systems[12–14]. During sleep, neurons are spontaneously active without external input and generate complex patterns of synchronized activity across brain regions[15,16]. Two critical components which are believed to underlie memory consolidation during sleep are spontaneous replay of memory traces and local unsupervised synaptic plasticity[17–19]. Replay of recently learned memories along with relevant old memories[20–25] enables the network to form orthogonal memory representations to enable coexistence of competing memories within overlapping populations of neurons[19,26–28]. Local plasticity allows synaptic changes to affect only relevant memories. While consolidation of declarative memories presumably depends on the interplay between fast-learning hippocampus and slow-learning cortex[20] ('Complementary Learning Systems Theory'), several types of procedural memories (e.g., skills) are believed to be hippocampus-independent and still require consolidation during sleep, particularly during Rapid Eye Movement (REM) sleep[29]. These results from neuroscience suggest that sleep replay principles applied to ANNs may reduce catastrophic forgetting

[1]Neurosciences Graduate Program, University of California San Diego, La Jolla, CA 92093, USA. [2]Department of Medicine, University of California San Diego, La Jolla, CA 92093, USA. [3]Department of Computer Science, New Mexico Tech, Soccoro, NM 87801, USA. ✉e-mail: mbazhenov@ucsd.edu

in machine learning models. In this new study, we focus on the hippocampus-independent consolidation of memories during REM sleep-like activity.

We show that implementing a sleep-like phase after an ANN learns a new task enables replay and makes possible continual learning of multiple tasks without forgetting. These results are formalized as a Sleep Replay Consolidation (SRC) algorithm as follows. First, an ANN is trained using the backpropagation algorithm, denoted below as awake training. Next, spontaneous brain dynamics, similar to those found in sleep[16,30], are simulated and we run one-time step of network simulation propagating spontaneous activity forward through the network. Next, we do a backward pass through the network in order to apply local Hebbian plasticity rules to modify weights. After running multiple steps of this unsupervised training phase, testing or further training using regular backpropagation is performed. SRC can be applied alone or in combination with state-of-the-art rehearsal methods to further improve these methods' performance. We recently found that this approach can promote domain generalization and improve robustness against adversarial attacks[31]. Here, we developed the sleep algorithm to address continual learning problem. We show that spontaneous reactivation of neurons during sleep engages local plasticity rules that can recover performance on tasks that were thought to be lost due to catastrophic forgetting after new task training.

## Results

### Sleep Replay Consolidation (SRC) algorithm implementation and testing strategy

In animals and humans, spontaneous neuronal activity during sleep correlates with that during awake learning[12]. This phenomenon, called sleep replay, along with Hebbian plasticity, plays a role in strengthening important and pruning irrelevant synaptic connections underlying sleep-dependent memory consolidation[20,32]. To integrate the effect of sleep into artificial neuronal systems, we interleaved incremental ANN training using backpropagation with periods of simulated sleep-like activity based on local unsupervised plasticity rules (see Methods for details). With this approach we were able to combine the "best of both worlds"—state of the art training performance delivered by modern deep neural network architectures[33] and important properties of biological sleep, including local plasticity and spontaneous reactivation[20]. Importantly, our approach is fully executed within a standard machine learning environment, and the sleep function can be easily added to any ANN type and any training algorithm.

In order to implement a sleep replay phase with local plasticity rules (referred to as the Sleep Replay Consolidation (SRC) algorithm below), the network's activation function was replaced by a Heaviside function (to mimic spike-based communications that occur in the brain) and network weights were scaled by the maximum activation in each layer observed during last training, in order to increase activity during the sleep phase. The scaling factor was determined based on pre-existing algorithms developed to run trained ANNs on neuromorphic hardware, such as spiking neural networks[34], while Heaviside-neuronal spiking thresholds were determined based on a hyperparameter search. To modify network connectivity during sleep phase we used an unsupervised, simplified Hebbian-type learning rule, which was implemented as follows: synaptic weights between two neurons are increased when both pre- and post-synaptic neurons are activated sequentially; synaptic weights are decreased between two neurons when the post-synaptic node is activated but the pre-synaptic node is silent (does not reach activation threshold). Further, to ensure sufficient network activity during the sleep phase, the input layer of the network was activated with noisy binary inputs. In each input vector at each time step of SRC, the probability of assigning a value of 1 to a given input pixel is taken from a Poisson distribution with mean rate calculated as the mean intensity of that input element across all the inputs observed during all of the preceding training sessions. Thus,

e.g., a pixel that was typically bright in all training inputs would be assigned a value of 1 more often than a pixel with lower mean intensity. Therefore, the only old task information that needs to be stored for future SRC applications is the mean input layer activation across all the past tasks and this information does not scale with the number of tasks. Importantly, no inputs representing specific memories were ever presented to the network during sleep; the state of the network (weight matrices) implicitly determined the patterns of reactivation and, ultimately, what was replayed during sleep.

In this study we analyzed SRC in the context of both class-incremental learning and cross-modal tasks. Class-incremental learning occurs when a network learns a series of classes (e.g., MNIST digits) incrementally without access to previously learned classes. In this case, performance is measured as the network's ability to classify and distinguish all classes. Cross-modal tasks measure the network's ability to store two distinct tasks (e.g., MNIST digits and Fashion MNIST images) in the same parameter space. We first utilized a toy example of binary patterns to analyze how synaptic weights change during sleep to support incremental learning. We then tested SRC in an incremental learning framework on the MNIST, Fashion MNIST, CUB-200, and CIFAR10 datasets (see Methods). For cross-modal tasks, we measured the ability of an ANN to learn sequentially both the MNIST and Fashion MNIST datasets when the network could only access one dataset during training time.

### SRC promotes consolidation of overlapping binary patches

We first tested the sleep algorithm by training a small network with just an input and output layer to distinguish four binary $10 \times 10$ images (see Fig. 1A) presented sequentially as two tasks (Task 1—first two images; Task 2—second two images). The network was always tested on its ability to classify all four images using a softmax classifier with no momentum. The amount of interference was measured as the number of overlapping pixels between images. Catastrophic forgetting should not occur when there is no interference between the images but, as the number of overlapping pixels increases, new task training can lead to forgetting. Our studies using biophysical models of the thalamocortical network[27,28] revealed that catastrophic forgetting occurs because the network connectivity becomes dominated by the most recently learned task, so input patterns for old tasks are insufficient to activate their respective output neurons; sleep replay can redistribute the network resources more equally between tasks to allow correct recall of the old and new memories.

After training on the first two images (denoted T1), the network could classify T1 images accurately but has not yet learned the other two images, so overall performance was 50% (Fig. 1B, dashed blue line). Here, when, e.g., the first image was presented to the network, input to its corresponding output neuron was greater than the inputs to the other output neurons (Fig. 1C, left group of bars, 12 pixel overlap)—correct classification. After training on the second two images (denoted T2), the network either learned them without interference to T1 (performance increased to 100%—all four images are classified correctly), when there was little overlap between tasks (Fig. 1B, dashed red line, overlap is less than 12 pixels), or forgot the first two images (performance remained at 50%—T1 is erased and everything is classified as T2), when there was large overlap between tasks (Fig. 1B, dashed red line, overlap is more than 12 pixels). In the last case, presenting the first image resulted in greater activation of T2 output neurons (Fig. 1C, red bar in middle group) than T1 output neurons (Fig. 1C, black bar in middle group)—catastrophic forgetting. When SRC was applied following T2 (Fig. 1B, yellow line), T1 was recovered even for large overlaps between tasks (compare yellow and red lines for overlaps in range 12–16 pixels in Fig. 1B). Here, we observed that although input to the T1 output neuron (upon first image presentation) remained unchanged (Fig. 1C, black bar in rightmost group), the input to the T2 output neurons decreased following SRC (Fig. 1C, red bar in rightmost group).

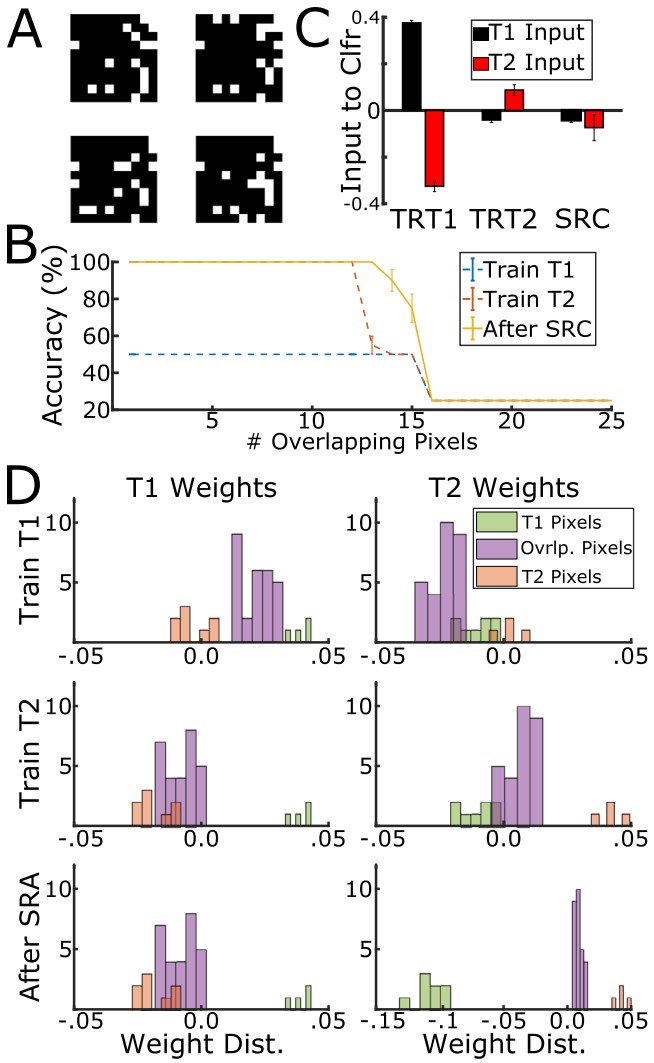

**Fig. 1 | SRC reduces catastrophic forgetting for sequential task training.**
**A** Example of four binary images with 15 pixel overlap divided into task 1 (T1, top two images: #0,1) and task 2 (T2, bottom two images: #2,3). T2 was trained after T1. **B** Classification accuracy as a function of number of overlapping pixels (inter- ference) after training T1 (blue), T2 (red), and SRC (yellow). The network was always tested for all 4 images. Note, that performance is significantly higher after SRC in the range of pixel overlap 12–16 (error bars indicate standard error of the mean). **C** Input to the T1 output neuron 0 (black) and max input to the T2 output neurons 2 and 3 (red) when presented with T1 image 0 after subsequent stages of training (T1 = TRT1, T2 = TRT2) and sleep (SRC). **D** Distribution of weights connecting to T1 (left) or T2 (right) output neurons. Color is based on category of connections: Unique—connections from neurons representing pixels unique for T1 or T2; Overlapping—connections from neurons representing overlapping pixels between T1 and T2. Rows show subsequent stages of training and sleep.

Thus, after SRC, the network was able to withstand larger amounts of interference, indicating that SRC is beneficial in reducing catastrophic interference for this simplified task.

We next examined the synaptic weight changes after training the second task, and after SRC. Figure 1D shows histograms of synaptic weights from input layer neurons to four output neurons (T1 neurons on the left and T2 neurons on the right) for the 14-pixel overlap con- dition, which results in catastrophic forgetting following T2 training and complete recovery following SRC. We separated weights from input neurons representing uniquely T1 and T2 pixels (Fig. 1D, green, orange), as well as overlapping pixels between T1 and T2 (purple). After T1 training (Fig. 1D, top row), the weights from T1 pixels (both unique and overlapping) to T1 decision neurons increased (purple and green

distributions on the left), while connections to T2 decision neurons became negative (purple and green on the right). After T2 training (Fig. 1D, middle row), the weights from T2 pixels (both unique and overlapping) to T1 output neurons decreased (purple and orange on the left), while connection to T2 output neurons increased and became positive (purple and orange on the right). Strong input to T2 output neurons from T1 overlapping pixels (purple on the right) overcame input to T1 output neurons from unique T1 pixels (green on the left, see also Fig. 1C, middle group). This resulted in catastrophic forgetting of T1, as T1 inputs led to higher activation of T2 output neurons. After SRC (Fig. 1D, bottom), weights from T1 unique pixels to T2 output neurons became inhibitory (green distribution on the right) while most other categories of weights remained unchanged. Thus, before SRC, pre- senting T1 images resulted in preferential activation of T2 output neurons and misclassification. After SRC the same T1 inputs inhibited T2 output neurons, so T1 output neurons displayed relatively higher activation leading to correct classification.

Since T1 neurons did not spike during sleep in this simple model, the weights to T1 output neurons did not change (compare Fig. 1D left, middle vs bottom row); however, for more complex tasks and network architectures, all the weights could change after sleep phase (see below). This simple model analysis revealed that SRC down-scales synaptic weights from task-irrelevant neurons, thereby reducing cross- talk between tasks. A more rigorous analysis of this toy model revealed that following sequential training on T1 and T2, the network weights still had positive cosine similarity to the weights of the network trained on T1 alone (See Supplementary Information, Section 1). This demonstrates that even when catastrophic forgetting is observed from classification perspective, the network weights may still preserve information about previous tasks.

## SRC algorithm reduces catastrophic forgetting on standard datasets

ANNs have been shown to suffer from catastrophic forgetting for various standard image datasets including MNIST, CUB-200, and CIFAR-10[35]. To test SRC for these datasets, we created 5 tasks (per dataset) for the MNIST, Fashion MNIST, and CIFAR-10 datasets and 2 tasks for the CUB200 dataset. Each pair of items in the MNIST (e.g., digits 0 and 1), Fashion MNIST and CIFAR-10 datasets was defined as a single task, and half of the classes in CUB200 was considered a single task. Tasks were trained sequentially and each new task training was followed by a sleep phase until all tasks were trained. This mimics interleaving periods of awake training with periods of sleep in the biological brain.

A baseline ANN with two hidden layers (see Methods for details) trained incrementally without sleep suffered from catastrophic for- getting, representing the lower bound on performance (Table 1, Sequential Training). The ideal accuracy of the same network trained on all tasks at once represents the upper bound (Table 1, Parallel Training). We found a significant improvement in the overall perfor- mance compared to the lower bound (Table 1, SRC vs Sequential Training), as well as task-specific performance (Fig. 2) when SRC was incorporated into the training cycle. On CUB-200, the baseline ANN suffered from catastrophic forgetting after it was trained sequentially on two tasks (first task–5%, second task–95%). Incorporating SRC after each task training resulted in much higher and balanced classification accuracy (first task–63.2%, second task–45.4%). Similar results were found for CIFAR-10, where the network implementing SRC achieved overall accuracy values of 44%, significantly higher than the control ANN without SRC (19%). Errors in Table 1 represent the standard deviation across 5 trials with different network initialization and dif- ferent task orders. Note that computational costs for running SRC are comparable with the costs of training each additional task (when task training is implemented in batches). However, sleep required much less inputs to pass through the network; thus, the computational

**Table 1 | Average test accuracy (± standard deviation; performance is averaged across different task orders) for baseline sequential training, Elastic Weight Consolidation, Synaptic Intelligence, Orthogonal Weight Modification, SRC algorithm, Rehearsal with 0.75% of old data stored, SRC algorithm + Rehearsal (0.75% of old data stored), and the ideal performance of a network trained on all data at once**

| Method | Inc. MNIST | Inc. Fashion MNIST | Multi-modal MNIST | Inc. CUB-200 | Inc. CIFAR10 |
|---|---|---|---|---|---|
| Sequential training | 19.49 ± 0.002 | 19.67 ± 0.003 | 47.18 ± 0.0020 | 5.32, 95.41 | 19.01 ± 0.002 |
| EWC | 20.37 ± 0.24 | 21.39 ± 2.93 | 74.55 ± 0.83 | 0.0, 63.85 | 18.54 ± 0.39 |
| SI | 21.38 ± 1.34 | 22.05 ± 2.05 | 74.15 ± 0.80 | 0.07, 60.01 | 22.84 ± 0.129 |
| OWM | 77.038 ± 2.91 | 58.35 ± 2.05 | 91.29 ± 1.05 | 71.4, 21.5 | 34.234 ± 1.87 |
| SRC | 48.47 ± 5.03 | 41.68 ± 5.04 | 61.33 ± 0.0150 | 63.2, 45.4 | 44.55 ± 1.45 |
| Rehearsal (0.75% data) | 79.91 ± 5.34 | 55.192 ± 7.74 | 83.13 ± 0.89 | 42.32, 51.49 | 39.39 ± 0.64 |
| Rehearsal (0.75% data) + SRC | 86.47 ± 1.061 | 67.818 ± 3.64 | 83.18 ± 1.91 | 56.55, 38.05 | 58.24 ± 0.561 |
| Parallel training | 98.02 ± 0.006 | 87.86 ± 0.005 | 90.05 ± 0.0028 | 85.49, 79.15 | 72.43 ± 0.002 |

Ten batches of training were used to ensure a complete convergence of the training method. For CUB-200, we report task 1 performance and task 2 performance separately.

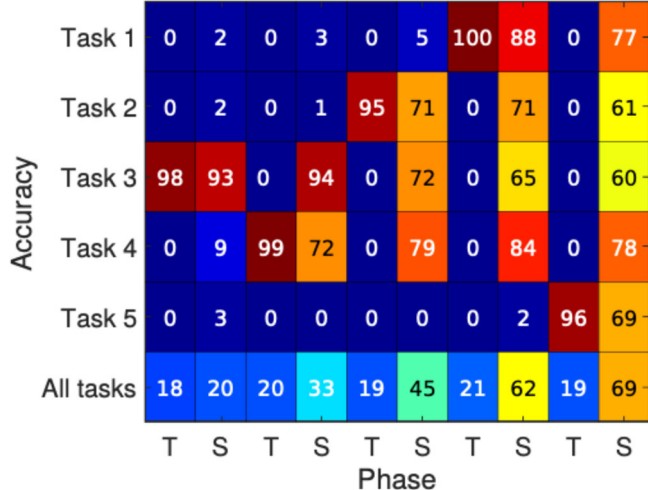

**Fig. 2 | Interleaving new task training with sleep results in recovery of old tasks' performance on MNIST dataset.** Sleep phase (S) was implemented after each new task training (T). Task 1–0/1, Task 2–2/3, Task 3–4/5, Task 4–6/7, Task 5–8/9. Each column shows performance for all tasks after either new task training or sleep as labeled below.

performance of sleep phase can be likely improved by incorporating the idea of mini-batches (see Supplementary Information Fig. 2 for more detail).

We next tested the SRC algorithm using a cross-modal task, where the network first learned the MNIST and next the Fashion MNIST dataset. An ideal network trained on both datasets at once achieved a classification accuracy of around 90%. When trained incrementally, the baseline ANN failed to accurately classify the MNIST data, achieving overall classification accuracy of 47% (Table 1). Incorporating SRC into the training boosted overall classification accuracy to 61%. While we primarily tested a network with only two hidden layers, the analysis of 4-hidden layer networks on MNIST task revealed that SRC can be applied in deeper architectures to recover old tasks' performance.

Although we report here SRC performance numbers lower than those for some generative models[36,37], SRC was able to reduce catastrophic forgetting with only limited (average statistics) knowledge of previously learned examples and solely by utilizing spontaneous replay driven by the weights important for representation of old tasks. Among methods operating without access to old data, SRC surpassed regularization methods, such as Elastic Weight Consolidation (Table 1, EWC) and Synaptic Intelligence (Table 1, SI)[38,39], on all class-incremental learning tasks and revealed reduced performance only for the multi-

modal task (see Discussion and Supplementary Information). However, a recently developed regularization method (Orthogonal Weight Modification, OWM) slightly surpasses performance of SRC on most tasks, suggesting that regularization methods can successfully promote recovery of old tasks in a class-incremental learning setting[40]. One implementation advantage of SRC over OWM is that SRC is an offline method (just like biological sleep), so it can be run after the normal training process is completed. Therefore, SRC can be directly combined with the state-of-the-art rehearsal methods for continual learning to further improve these methods' performance and/or reduce amount of old data they need to store (see below). For tasks where it is unknown a priori if/when new training would be needed, SRC can be easily applied post fact, as only past average input would be needed to run SRC.

Ultimately, our results suggest that information about old tasks is not completely lost even when catastrophic forgetting is observed from the performance-level perspective. Instead, information about old tasks remains present in the synaptic weights and can be resurrected by offline processing, such as sleep replay. Biological sleep is a complex phenomenon and our implementation of sleep replay is likely oversimplified. Future studies, can improve SRC implementation (e.g., by combining REM-type and NREM-type sleep) to further improve performance.

## SRC is complementary to state-of-the-art rehearsal methods
Many generative solutions aimed at solving catastrophic forgetting train a separate generator network to recreate and make use of the old data during new training sessions—commonly called replay—to prevent forgetting[36,37]. We next tested the complementary effect of incorporating old training data during new training sessions along with SRC. In this scenario, we included a small percentage of old task data during new task training and modified the loss function to promote balanced learning of both new and old tasks (see Methods and Supplementary Information). The small amount of old task examples alone (without SRC) resulted in an increase in network classification accuracy on most tasks, compared to a sequentially trained network (Table 1, Rehearsal, Fig. 3, black lines). When SRC was run after rehearsal (Table 1, Rehearsal+SRC, Fig. 3, red lines), this significantly boosted the overall classification accuracy even further when compared to using a small fraction of old data alone. These results suggest that SRC can reduce the amount of data needed to be generated or stored with state-of-the-art rehearsal methods, while still obtaining near ideal accuracy. Rehearsal benefits from longer training time (see Supplementary Information, Fig. 7). Our results suggest that SRC could support rehearsal methods by reducing the training time in addition to reducing memory capacity requirements. This point is further explored below.

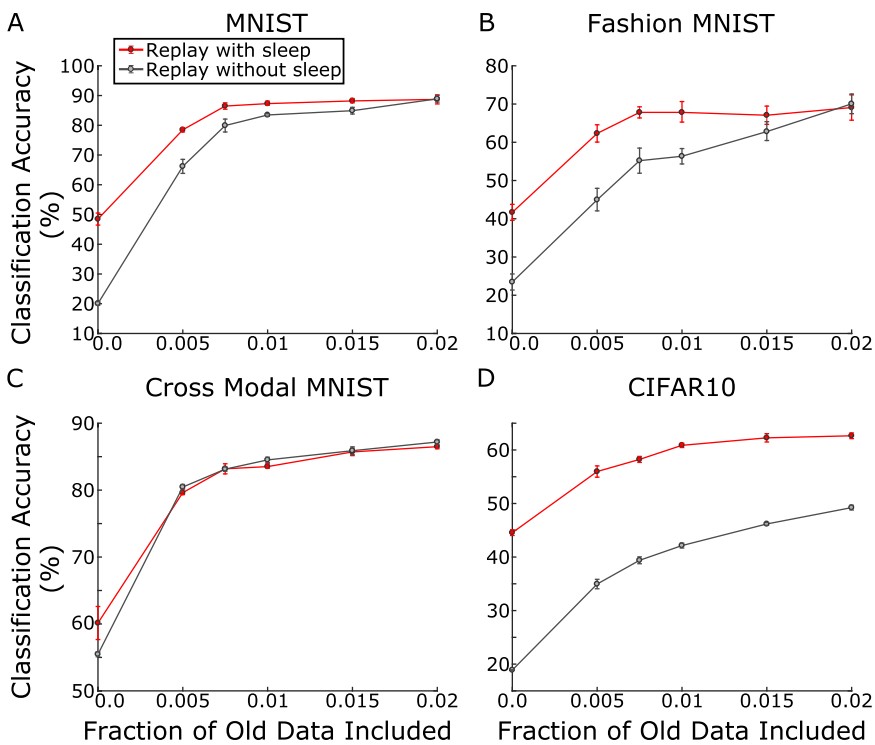

**Fig. 3 | SRC is complementary to rehearsal - replay methods which store/generate old examples and combine new data with these examples during training.** The classification accuracy as a function of percent of old data included is shown for **A** MNIST, **B** Fashion MNIST, **C** Cross Modal MNIST, and **D** CIFAR10. Error bars represent standard errors across 5 training sessions with different task orders.

From a neuroscience standpoint, this result predicts that learning new memories first by a separate, such as hippocampal, network, that stores not only new memories but also some high-level information about old memories ("indexes"[41]), and subsequently training another, such as cortical, network using output of the hippocampal network, followed by pure cortical replay, gives an optimal solution to embed new memories to cortex and to protect old memories. This is indeed how we believe brain learns hippocampus dependent declarative memory tasks during slow-wave sleep[20].

Next, we discuss in more details how incorporating SRC on top of a state-of-the-art rehearsal method, iCaRL, can result in better continual learning performance than iCaRL alone (details on iCaRL implementation can be found in Supplementary Information or the original paper[42]). Table 2 shows performance when different memory capacities ($K$, number of stored examples from previous classes) were used. In the case of MNIST, iCaRL with K = 100 achieved a performance of 65.5% after 10 epochs/task and iCaRL with K = 200 achieved a performance of 76.9% after 10 epochs/task. iCaRL+SRC with $K$ = 100 ($K$ = 200) achieved a classification performance of 78.1% (84.5%). Thus, a higher accuracy can be obtained by combining iCaRL with SRC and it may even be achieved with a lower memory capacity. For MNIST, Fashion MNIST and CIFAR10 datasets, in almost all cases (except $K$ = 2000 for MNIST) iCaRL + SRC had higher performance than iCaRL alone for the same memory capacity. Previously, we reported that OWM could surpass performance of SRC alone. However, the performance of iCaRL + SRC with $K$ = 100 always exceeded OWM performance, suggesting that rehearsal methods are still the state-of-the-art in class-incremental continual learning settings and combining them with SRC may deliver the most accurate solution for a given memory capacity.

In addition to lowering memory requirements, SRC could also reduce training time (denoted as a number of epochs per task) needed to achieve optimal results. Indeed, we found that iCaRL + SRC converges more rapidly than iCaRL alone. For example, for $K$ = 50, iCaRL + SRC achieved after just one epoch/task of training the same accuracy as iCaRL alone after 10 training epochs/task (see Supplementary Information Fig. 7). For $K$ = 100, only 4 epochs/task were required with iCaRL + SRC before the same accuracy was obtained after 10 epochs/task with iCaRL alone. For $K$ = 1000, iCaRL + SRC after 8 epochs/task had a final accuracy of 87.7%, whereas iCaRL alone even after 10 epochs/task only achieved a final accuracy of 87.32%. In general, the benefits of SRC were higher for lower values of K. We defined the training savings as the number of epochs/task after which iCaRL + SRC achieves a greater performance than iCaRL alone after 10 epochs/task. The training savings on all 3 datasets (averaged across all memory capacities and task orders) were: 3.73 epochs/task for MNIST, 3.67 epochs/task for Fashion MNIST, and 2.80 epochs/task for CIFAR10 (see Supplementary Information Fig. 7 for example plots).

In sum, we found that the SRC algorithm can support various rehearsal methods by (a) reducing the amount of old data that are stored (or allowing only replay of the highest-quality generated examples); (b) reducing training time.

## SRC reduces catastrophic forgetting by replaying old task activity

How does SRC work? From a neuroscience perspective, sleep reduces interference by replaying activity of recently learned tasks and old relevant (interfering) tasks[20]. Using biophysical models of brain network and testing them for simplified task of learning overlapping memory sequences, we showed that sleep replay modifies the synaptic weights to create unique synaptic representation for each task[28]. Such differential allocation of resources leads to reduced representational overlap and therefore diminishes catastrophic forgetting.

To address how SRC works for ANNs, we examined a reduced class-incremental learning MNIST task. The network was first trained on digits 0 and 1 (Task 1), followed by sleep. Next, the network was trained on digits 2 and 3 (Task 2), leading to Task 1 forgetting, followed by second sleep phase. We then analyzed the second period of sleep

**Table 2 | SRC improves upon iCaRL method with different memory capacities**

|  | K = 50 | K = 100 | K = 200 | K = 500 | K = 1000 | K = 2000 |
|---|---|---|---|---|---|---|
| iCaRL, MNIST | 53.428 ± 5.20 | 65.502 ± 4.66 | 76.856 ± 5.29 | 87.326 ± 1.30 | 90.628 ± 0.41 | 92.850 ± 0.44 |
| SRC + iCaRL, MNIST | 69.97 ± 3.74 | 78.086 ± 3.16 | 84.498 ± 1.39 | 88.862 ± 0.44 | 91.130 ± 0.58 | 92.742 ± 0.41 |
| iCaRL, Fashion MNIST | 49.342 ± 6.83 | 57.786 ± 3.13 | 62.828 ± 2.97 | 69.038 ± 2.78 | 73.972 ± 1.58 | 78.030 ± 0.62 |
| SRC + iCaRL, Fashion MNIST | 51.554 ± 11.63 | 61.916 ± 5.25 | 65.110 ± 2.95 | 69.798 ± 2.48 | 75.226 ± 1.28 | 78.542 ± 1.17 |
| iCaRL, CIFAR10 | 35.156 ± 3.41 | 43.244 ± 1.99 | 49.102 ± 2.07 | 54.898 ± 1.49 | 59.528 ± 0.77 | 62.878 ± 0.65 |
| SRC + iCaRL, CIFAR10 | 39.382 ± 3.61 | 46.002 ± 2.07 | 51.324 ± 1.86 | 57.504 ± 0.61 | 61.23 ± 0.88 | 64.018 ± 0.56 |

Standard deviations reported across 5 randomly initialized networks and task orders.

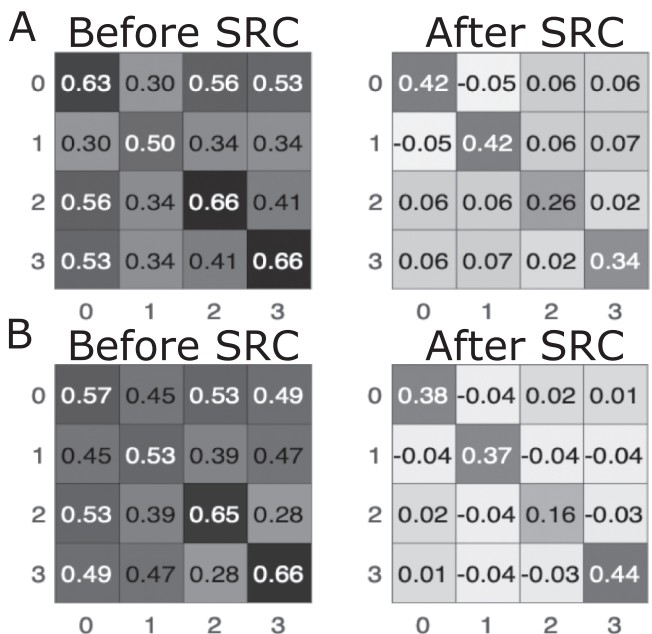

**Fig. 4 | SRC reduces correlations between image classes while maintaining strong correlations within classes.** Correlation matrices of activations in hidden layer 1 (**A**) and layer 2 (**B**). Labels (0–3) indicate image class: Task 1–0/1, Task 2–2/3. Note that before SRC (left) correlations between classes, e.g., images of 1 and 2, are almost as high as correlations within classes, e.g., different images of 1.

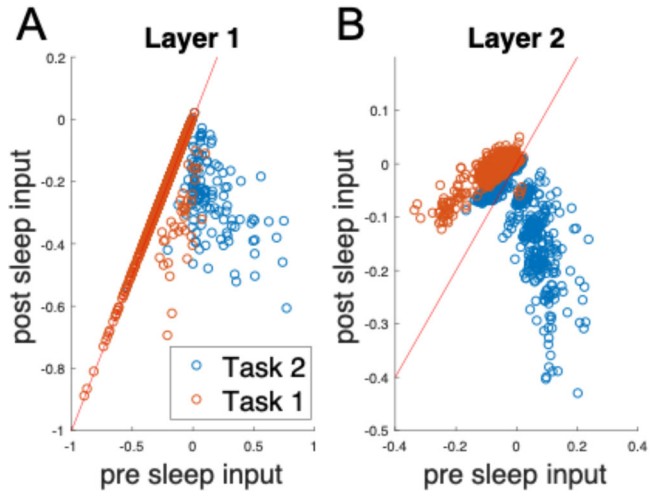

**Fig. 5 | SRC changes input to the hidden layer neurons to favor old tasks. A** In layer 1 (left), input to the neurons representing old task is reduced less than input to the neurons representing most recently learned task. **B** In layer 2 (right), SRC increases input to the neurons representing old task while decreasing input to the neurons representing most recently learned task.

that resulted in recovery of performance on the first task (overall accuracy on both tasks after SRC = 90%).

To test if after SRC the neurons are differentially allocated to represent digits from different tasks, we first looked at the correlation of activities in the hidden layers. The ANN was presented with all inputs from the test sets of both tasks and we calculated the average correlation within the first and second hidden layers before and after application of SRC (Fig. 4). In both layers, before SRC (when all inputs were classified as either 2 or 3), correlations of activity for digits from different classes were almost as high as correlations of activity for digits from the same class (with exception of classes 2 and 3 in the 2nd hidden layer) (Fig. 4B, left). After SRC we observed decorrelation (near-zero correlations between representation of digits from different classes) and an increase in representational sparseness (where each stimulus strongly activates only a small subset of neurons) for all four digits (Fig. 4, right and Supplementary Information Fig. 9). This suggests that SRC prevents interference between classes by allocating different neurons to different tasks, thereby creating a distinct population code for different input classes.

Decorrelation of activity alone may not explain the old task recovery. Indeed, if activity of neurons representing the earlier task remains lower than that for the later task, then catastrophic forgetting would still occur. Therefore, we next analyzed activation properties of

the neurons representing individual tasks (Fig. 5). In each hidden layer, we selected the top 100 neurons that were most active in response to Task 1 or Task 2 inputs (see Methods for details on selection process) and analyzed how input to these neurons changes after SRC. In the first hidden layer, both Task 1 and Task 2 neurons experienced a decrease in input strength, but the effect was generally stronger for Task 2 neurons, reflecting a greater decrease in the weights connecting to Task 2 neurons (Fig. 5, left). More notably, in the second hidden layer, we observed an increase of the input to the Task 1 neurons but a decrease of the input to the Task 2 neurons (Fig. 5, right). This suggests that a relative increase in activity of Task 1 neurons along with an overall decorrelation of representations between the tasks explains recovery of performance on the old task.

In the biological brain, sleep-dependent memory consolidation occurs through memory replay, i.e., patterns of neurons activated during task learning are reactivated spontaneously during sleep[12]. To test if replay happens during SRC, we looked at the firing activity of Task 1 and Task 2 neurons during sleep. To avoid possible bias from using task averaged input, here the network was stimulated during sleep by a completely random input. We calculated the average firing rates of neurons and we found in the first hidden layer that the top 100 most active neurons involved in representation of the previously learned Task 1 and Task 2 digits had a higher average firing rate during sleep than a randomly selected subset of neurons (Fig. 6A). In the second layer, this result was more pronounced for the most recently learned task (digits 2 and 3) (Fig. 6B). To compare firing rates of digit-specific neurons, we concatenated the firing rates of digit-2 and digit-3 (Task 2) neurons and compared them with the concatenated firing

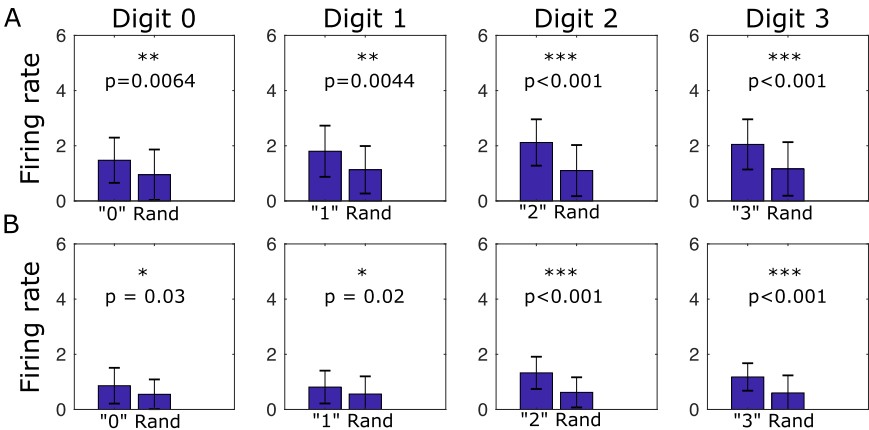

**Fig. 6 | Task-specific neurons are more likely to be activated during sleep.** Spiking activity of digit-specific neurons during sleep is greater than activity of a random subset of neurons in both first (**A**) and second (**B**) hidden layers. Neurons are activated by random (not averaged) inputs to the input layer. Error bars represent standard deviation. *p*-values computed from two-sided *t*-test comparing firing rates of digit-specific neurons vs. random subset of neurons with Bonferroni Correction.

rates of digit-0 and digit-1 (Task 1) neurons. We found that digit-2 neurons and digit-3 neurons were significantly more active in layer one (t(200) = 3.456, *p* = 0.004, one-sided *t*-test, Bonferroni Correction) and layer two (t(200) = 5.215, *p* < 0.001, one-sided *t*-test, Bonferroni Correction). Together, this suggests that spontaneous firing patterns during sleep are correlated with activity observed during task learning, in agreement with neuroscience data[12,20]. Interestingly, sleep replay improved performance not only by increasing connectivity for the old tasks but also by reducing connectivity representing the most recent task.

If replay indeed provides a mechanism of how sleep protects memories from interference, then sleep replay should increase performance for a single task. Indeed, the most well-established neural effect of sleep on memory in the brain is augmenting memory traces for recent tasks[20]. In this new study we typically observed a reduction in performance on the most recent task. Do we have contradiction here? To test this, we tested effect of SRC on a single task memory performance as a function of the amount of initial training. We found (see Supplementary Information, Fig. 5) that when the network is undertrained, i.e., initial performance is low, SRC can greatly increase performance without involving any new training data. However, if the memory is well trained, SRC cannot improve or even slightly reduces performance. This is consistent with neuroscience data suggesting an inverse association between learning performance pre-sleep and gains in procedural skills post-sleep in humans, i.e., good learners exhibit smaller performance gains after sleep than poor learners[43].

## Discussion

We implemented an unsupervised sleep replay consolidation (SRC) algorithm for artificial neural networks and we showed that SRC can alleviate catastrophic forgetting for several different datasets, ranging from binary patterns to natural images. SRC simulates fundamental properties of a biological sleep replay, including: (a) spontaneous activity of neurons during sleep leads to replay of previously learned activation patterns, (b) unsupervised Hebbian-type plasticity modifies the connectivity matrix to increase task-specific connections and to prune excessive connections between neurons. This increases sparseness of representation and reduces representational overlap amongst distinct input classes. Our results suggest that unsupervised Hebbian-type plasticity combined with spontaneous activity to simulate sleep-like dynamics can help alleviate catastrophic forgetting in an incremental learning setting.

Existing approaches to prevent catastrophic forgetting generally fall under two categories: rehearsal and regularization methods[35]. Rehearsal methods combine previously learned data, either stored or

generated, with novel inputs in the next training to avoid forgetting[36,44–48]. This approach includes models where distinct networks, loosely representing hippocampus and cortex, are used to generate examples from a distribution of the previously learned tasks[37]. As the number of previously learned tasks grows, this approach would require increasingly complex generative networks capable of potentially generating everything that was learned before. In contrast, in a biological brain, sleep replay is spontaneous and occurs without any external input. Thus, although rehearsal methods work well from an engineering standpoint, they unlikely capture the mechanisms that nature developed to enable continual learning. While current SRC performance, as we report here, is somewhat inferior compared to the state-of-the-art rehearsal techniques[36,37,44], our approach does not require storing any task-specific information or training new generator networks. Furthermore, SRC can be complementary to rehearsal methods by incorporating partial old data replay, reducing the amount of old data needed for new task training. Importantly, SRC was also shown to perform other memory functions associated with sleep, such as promoting generalization and improving robustness against adversarial attacks[31]. While actual training data were used to generate replay to improve generalization in ref. 31, here we show that sleep replay can alleviate catastrophic forgetting on class-incremental learning task by only having access to the basic input statistics.

Regularization approaches[49] for reducing catastrophic forgetting aim to modify plasticity rules by incorporating additional constraints on gradient descent such that important weights from previously trained tasks are maintained. The Elastic Weight Consolidation (EWC) and Synaptic Intelligence (SI) methods penalize updates to weights deemed important for previous tasks[38,39]. Although these studies report positive results on preventing catastrophic forgetting in various tasks (MNIST permutation task, Split MNIST, Atari games), they may not work well in a class-incremental learning framework, where one class of inputs is learned at a time[36]. Since these approaches support continual learning by stabilizing weights that are deemed important for earlier tasks, high overlap in representation between tasks belonging to the same dataset may reduce their performance (see Supplementary Information). In terms of memory constraints, SRC is similar to regularization methods since only a fixed amount of information, independent of the number of tasks trained, is needed to be stored, but SRC exceeds performanace of EWC and SI approaches in an incremental learning framework that is natural for biological systems. Orthogonal Weight Modification (OWM) improves upon these methods by computing a projector matrix for old tasks and only allowing weight updates in an orthogonal direction to this projector[40].

OWM was shown to be more successful than EWC and SI on class-incremental learning tasks.

Why can a noise driven reactivation, as proposed in the SRC algorithm here, recover memories, including higher-order associations, learned during training? In biology, while the brain generates sleep activity spontaneously, the sleep firing patterns still reflect the synaptic weight structure that was learned during training. In fact, it was reported in vivo that similar spatial patterns of population activity were observed both when the neuron fired spontaneously and when it was driven by its optimal stimulus[50]. While the noisy input applied in the SRC algorithm may not include higher-order structure found in the training data, replay during sleep involves extracting such higher-order interactions, because the information is already present in the patterns of synaptic weights. Many results from biological sleep replay suggest that replay events are complex patterns, and not a simple repetition of the past inputs[51]. From that perspective, spontaneous replay found in the biological brain and implemented in the SRC algorithm here, is very different compared to explicit replay of past inputs often found in machine learning rehearsal methods (reviewed in refs. 52,53).

Our approach is implemented using a machine learning framework, where during the sleep phase we simply substitute in the Heaviside-activation function and modify the learning rule. Since the idea of SRC is inspired by biological spiking networks (SNNs), we also tested SNN-only implementation by performing "awake" training on MNIST data in an SNN via backpropagation[54] and found that incremental learning can lead to catastrophic forgetting. When sleep phase was implemented after each new task training using noise driven spontaneous replay, this recovered the old task performance (see Supplementary Information). While such framework may be suboptimal for ANNs, it has an advantage of avoiding explicit mapping ANN to SNN and may be efficient in the neuromorphic solutions where SNN is explicitly trained[55].

It remains an open question whether SRC would benefit convolution layers of the network, which were kept frozen in this study. Recent studies suggest that catastrophic forgetting does not occur in the feature extractor but takes place in the later layers, which motivated the use of SRC in these layers[56,57]. From a neuroscience perspective convolutional layers are somewhat equivalent to the primary visual cortex which is less plastic beyond development and unlikely to be involved in significant rewiring during sleep replay, in contrast, e.g., to the associative cortices.

Sleep replay helps to resurrect performance on tasks that were damaged after new training. This suggests that while ANN performance for old tasks is reduced, the network connectivity retains partial information about these tasks and spontaneous activity combined with unsupervised plasticity during sleep may reverse damage and reorganize connectivity to accommodate both tasks. Previously, it was shown that a wake-sleep algorithm developed for recurrent spiking neural networks, which does capture some principles of memory consolidation during sleep, can reduce the number of training examples needed to achieve optimal performance on single tasks[58]. The wake-sleep algorithm for Boltzmann Machines was shown to be able to learn representations of inputs and highlighted the role of incorporating a sleep phase to improve learning[59]. Our work further adds to this literature by exploring the role of sleep and local plasticity rules in overcoming catastrophic forgetting.

Our study makes several predictions for neuroscience: (a) Synaptic dynamics during sleep remain poorly understood. Some studies suggest net reductions of synaptic weights[32], while others argue for net increase[60]. Our work (extending ideas of ref. 28) predicts that sleep replay leads to complex reorganization of synaptic connectivity, including potentiation of some synapses and pruning of others with the goal of increasing separation between memories. We found that sleep replay may increase the contrast between memory traces by enhancing lateral inhibition, such that activation of one memory inhibits other similar memories to avoid interference[61]. Recent studies suggest that local learning rules can orthogonalize memories with different temporal contexts[62,63]. Our current (and recent ref. 28) work agrees with this idea and further suggests that local learning rules and sleep can orthogonalize representation of interfering memories, even with limited contextual information. (b) Sleep increases the sparseness of memory representations, which is in line with previous theoretical ideas about the role of interleaved training[1]. (c) Currently, sleep replay ideas are best developed for Non-Rapid Eye Movement (NREM) sleep when neuronal activity is structured by global network oscillations, such as sleep slow waves[64]. Our model predicts that sleep replay can be mediated by sparse patterns of excitation propagating through the network, which is typical for REM sleep (indeed, memory replay was also reported in REM sleep[65]). (d) Our model predicts the importance of neuromodulatory changes from learning (wake) to consolidation (sleep) phase, including strengthening of intracortical synapses and reduction of intrinsic excitability, as found in NREM sleep[66,67]. We suggest that these changes effectively increase the signal-to-noise ratio in network dynamics to promote the stronger memories, that are replayed and consolidated, while allowing forgetting of the weaker memory traces.

Catastrophic forgetting may be interpreted as asymmetry of the weight configuration that became biased towards the most recent task after new training. From this perspective, SRC may be seen as a "symmetry-correcting" mechanism. Indeed, Hebbian learning has been shown to orthogonalize neural codes and our recent biophysical modeling work[28], as well as results here, suggest that sleep orthogonalizes memory representations by "assigning" distinct weights to represent distinct memories. We found that rather than engage completion between memories, sleep allows chunks of competing memories to replay simultaneously and independently, so each memory would reach its optimal representation, which can be seen as a way of recovering the symmetry of weight distribution across tasks.

In sum, in this study we proposed an unsupervised sleep replay consolidation algorithm, inspired by the known role of biological sleep, to recover synaptic connectivity that otherwise would be forgotten after new training in ANNs. We tested a simplified model of sleep (noisy reactivation) and a Hebbian-type plasticity rule. Future work may need to explore more complex patterns of sleep activity (e.g., sleep waves) and learning rules (e.g., ref. 68), which could further improve performance.

## Methods

### Task protocols

To demonstrate catastrophic forgetting, we utilized an incremental learning framework, where a groups of classes are learned in a sequential fashion. We utilized 5 datasets to illustrate the prevalence of catastrophic forgetting as well as the beneficial role of sleep: a toy dataset termed "Patches", MNIST, Fashion MNIST, CUB-200, and CIFAR-10[69–72]. The Patches dataset consisted of binary patterns each belonging to its own class. The main advantage of this toy dataset is that it allows direct control over the amount of interference (number of overlapping pixels) in each of the binary patterns. This dataset was used to show the benefits of the sleep algorithm in a simpler setting and to reveal the exact weight changes during sleep replay, which led to a reduction in catastrophic forgetting. To ensure generalizability of our approach, we tested the sleep algorithm on the MNIST, Fashion MNIST, CIFAR10, and CUB-200 datasets. The MNIST and Fashion MNIST datasets are widely used in machine learning, consisting of 60,000 training images of hand-written digits or fashion items and 10,000 testing images.

CIFAR-10 is a similarly sized dataset with 10 classes of low-resolution natural images ranging from airplanes to frogs. For CIFAR-10, we used extracted features from a convolutional network with 3

**Table 3 | Neural network training parameters**

|            | MNIST/f. MNIST | CUB-200      | CIFAR-10     |
|------------|----------------|--------------|--------------|
| Arch. size | 1200, 1200, 10 | 350, 300, 200| 1028, 256, 10|
| Learn rate | 0.065          | 0.1,0.01     | 0.1          |
| Epochs/task| 10             | 50           | 10           |
| # class/task| 2             | 100          | 2            |
| Momentum   | 0.5            | 0.5          | 0.5          |
| Dropout    | 0.2            | 0.25         | 0.2          |

VGG blocks. The first block consists of 2 convolutional layers with 32 $3 \times 3$ filters in each layer, followed by a max pooling layer. The second block uses 2 convolutional layers with 64 $3 \times 3$ filters, followed by a max pooling layer. The last convolutional block consists of 2 convolutional layers with 128 $3 \times 3$ filters, again followed by a max pooling layer and a flattening operation. To train the convolutional model, we used two dense layers with 1028 and 256 units in each layer. For extraction, we took all inputs following the 3 convolutional blocks after the flattening layer, and used these input features to perform incremental learning. The original convolutional backbone was trained on the Tiny Imagenet dataset and CIFAR-10 images were fed through this network to extract intermediate feature representations[73]. The network was trained on Tiny Imagenet for 200 epochs using stochastic gradient descent with the following parameters: momentum = 0.9, learning rate = 0.005, batch size = 100, and the categorical cross-entropy loss function. CUB-200 contains natural images of 200 different bird species, with relatively few images per class. For CUB-200, following work done by ref. 37, we used the pre-trained Res-Net 50 embeddings. Here, the Res-Net 50 architecture was pre-trained on the ImageNet dataset[37,74].

The ANN was trained sequentially on five groups of two classes for MNIST, Fashion MNIST, and CIFAR-10 and 2 groups of 100 classes for CUB200, following previous studies[36]. In addition to testing the incremental MNIST and Fashion MNIST tasks (where two classes are learned during each task in training), we also tested the Multi-Modal MNIST task where first either MNIST or Fashion MNIST is learned and then during task 2, the other dataset is learned. This task tests the network's ability to develop representations of both datasets, digits and clothing, without catastrophic forgetting. After training on a single task, we run SRC algorithm as described below before training on the next task.

## Network details

Dataset specific parameters for training ANNs are shown in Table 3. This table includes the network architecture used to train the tasks in a sequential fashion (number of hidden units per layer), the learning rate used for each task, number of epochs per task, number of classes per task, as well as the dropout percentage used to train the network. The accuracy of the network on the entire dataset is listed in Table 1 in the main paper under Parallel Training. This denotes the accuracy of the network trained with the parameters listed in Table 3 but when the network has access to all training data during training time. Additionally, the Sequential Training row in Table 1 illustrates the performance of the ANN architecture when it is trained in an incremental fashion without the use of sleep or any other methods. These metrics serve both as an upper and lower bound, respectively.

The same network architecture and training parameters were used in all comparisons. For the MNIST, Fashion MNIST, and Multi-modal MNIST tasks, we used a fully connected architecture with two hidden layers consisting of 1200 nodes in each layer, followed by a classification layer with 10 output neurons. The network was trained for 10 epochs per task (two epochs in the case of Multi-modal MNIST), with mini-batch size of 100 images. For Multi-modal MNIST, the same network architecture was used, again with 10 output neurons. Thus, the same output neuron was used to represent both a digit and an article of clothing. For the CIFAR-10 dataset, we used extracted features from a convolutional network as denoted above. Extracted features were fed into a fully connected network with two hidden layers, with 1028 and 256 nodes in each hidden layer, respectively. These hidden layers also fed into a classification layer with 10 output neurons for each of the 10 classes in the dataset. This network was also trained for 10 epochs per task with mini-batch size of 100 images/feature vectors. For CUB-200, we used a network architecture consisting of two hidden layers with 350 and 300 nodes, connecting to a classification layer of 200 units, following work done by[37]. The network was trained for 50 epochs per task.

For all datasets, the ReLU nonlinear activation function was used during awake training in all layers. Each neuron in the network was trained without a bias term, which aids in the conversion to a spiking neural network (with Heaviside-activation function) during the sleep stage[34]. The networks were trained using the basic stochastic gradient descent optimizer with momentum. All networks were trained with the multi-class cross-entropy loss function. For incremental learning, this loss function was evaluated solely on the task being presented to the network during training time. Network weights were initialized using a random uniform distribution between −0.02 and 0.02 for MNIST, Fashion MNIST, Cross Modal MNIST and CUB-200, or −0.1 and 0.1 for CIFAR10. For comparisons between different methods, such as EWC and SI, the same training parameters and architectures were used. See Supplementary Information for more details on these regularization methods used to alleviate catastrophic forgetting.

## Sleep Replay Consolidation (SRC) algorithm

Here, we provide pseudocode (Algorithm 1) and more information about the Sleep Replay Consolidation algorithm (SRC) described in the main text. The intuition behind SRC is that a period of offline, noisy activity may reactivate network nodes that are responsible for representing earlier tasks. If network reactivation is combined with unsupervised plasticity, SRC will then strengthen the necessary and weaken unnecessary pathways through the network. If information about previously learned tasks is still present in the synaptic weight matrices, then SRC may be able to rescue apparently lost information. We start in the Main procedure, where first a network is initialized, e.g., within PyTorch or Matlab environment. Then, a task $t$ is presented to the network and the network is trained, as usual, via backpropagation and stochastic gradient descent. After supervised training phase, SRC is implemented within the same environment. During the SRC phase, the network's activation function is replaced by a Heaviside function and weights are scaled by the maximum activation in each layer observed during last training. The scaling factor and layer-wide Heaviside-activation thresholds are determined based on a pre-existing algorithm aimed at ensuring the network maintains reasonable firing activity in each layer[34]. This algorithm applies a scaling factor to each layer based on the maximum input to that layer and the maximum weight in that layer. During the SRC phase, we start with a forward pass, when the noisy input is created and fed through the network in order to get activity (spiking behavior) of all layers. Following the forward pass, a backward pass is run to update synaptic weights. To modify network connectivity during sleep we use an unsupervised simplified Hebbian-type learning rule, which is implemented as following: a weight is increased between two nodes when both pre- and post-synaptic nodes are activated (i.e., input exceeds Heaviside-activation function threshold); and a weight is decreased between two nodes when the post-synaptic node is activated but the pre-synaptic node is not (in this case, another pre-synaptic node is responsible for activity in the post-synaptic node). After running multiple steps of this unsupervised training during sleep, the final weights are rescaled again (simply by removing the original scaling factor), the Heaviside-type activation function is replaced by ReLU, and testing or further supervised training

on new data is performed. This all is implemented by a simple SRC function call after each new task training. Code is available on Github with the exact parameters dictating neuronal firing thresholds and synaptic scaling factors for each dataset and each architecture. These parameters were determined using a genetic algorithm aimed at maximizing performance on the training set. In the future, we would like to optimize these parameters based on ideal neuronal firing rates observed during sleep.

## Stimulation during sleep phase

During sleep phase, to ensure network activity, the input layer of the network is activated with noisy input (on/off pixels randomly assigned across the input layer). In each input vector (i.e., for each forward SRC pass), the probability of assigning a value of 1 (bright or spiking) to a given element (input pixel) is taken from a Poisson distribution with mean rate calculated as a mean intensity of that input element across all the inputs observed during all of the preceding training sessions. Thus, e.g., a pixel that was typically bright in all training inputs would be assigned a value of 1 more often than a pixel with lower mean intensity. Alternatively, the mean rate of the Poisson distribution used to create inputs may be chosen independently on the past ANN activation which still leads to the partial recovery of the old tasks.

## Algorithm 1. Sleep:

1: **procedure** ANNtoSleepANN(*nn*)
2: Change ReLU activation in ANN to Heaviside function in SleepANN and determine layer-wide specific threshold
3: Apply weight normalization and return scale, threshold for each layer **return** *SleepANN, scales, threshold*
4: **procedure** SleepANNtoANN(*nn*)
5: Directly map the new, unscaled weights from Heaviside-network (SleepANN) to ReLU network (ANN) **return** *ann*
6: **procedure** Sleep(*nn, I, scales, thresholds*) ▷ *I* is input
7: Initialize $v$ (voltage) = 0 vectors for all neurons
8: **for** $t \leftarrow 1$ to $Ts$ **do** ▷ $Ts$ - duration of sleep
9: $\mathbf{S}(1) \leftarrow$ Convert input $I$ to Poisson - distributed spiking activity
10: $\mathbf{S}$ = ForwardPass($\mathbf{S}, v, \mathbf{W}, scales, thresholds$)
11: $\mathbf{W}$ = BackwardPass($\mathbf{S}, \mathbf{W}$)
12: **procedure** ForwardPass($S, v, W, scales, threshold$)
13: **for** $l \leftarrow 2$ to $n$ **do** ▷ n - number of layers
14: $v(l) \leftarrow v(l) + (scales(l-1)\mathbf{W}(l, l-1)\mathbf{S}(l-1))$ ▷ W(l, l-1) - weights
15: $\mathbf{S}(l) \leftarrow v(l) > threshold(l)$ ▷ Propagate spikes
16: $v(l)(v(l) > threshold(l)) = 0$ ▷ Reset spiking neurons' voltages
**return** $S$
17: **procedure** Backward Pass($S, W$)
18: **for** $l \leftarrow 2$ to $n$ **do** ▷ n - number of layers
19: $\mathbf{W}(l, l-1) \leftarrow \begin{cases} \mathbf{W}(l, l-1) + inc & \text{if } \mathbf{S}(l, t) = 1 \& \mathbf{S}(l-1, t) = 1 \\ \mathbf{W}(l, l-1) - dec & \text{if } \mathbf{S}(l, t) = 1 \& \mathbf{S}(l-1, t) = 0 \end{cases}$ ▷ STDP
**return** $W$
20: **procedure** Main
21: Initialize neural network (*ANN*) with ReLU neurons and bias = 0.
22: **for** task $t = 1 : T$ **do**
23: Train *ANN* using backpropagation on task $t$.
24: *SleepANN, scales, thresholds* = ANNtoSleepANN(*ANN*)
25: *SleepANN* = Sleep(*SleepANN*, Training data $X$, *scales, thresholds*)
26: *ANN* = SleepANNtoANN(*SleepANN*)

## Including old data during training

In addition to testing the basic SRC when only mean input activation across all previous tasks is saved, we also tested how SRC can be complementary to existing state-of-the-art generative/rehearsal methods. To test this, we performed two additional experiments: one denoted rehearsal and another based on a near state-of-the-art method, iCaRL. The reherasal method included a percentage of old task data during new task learning sessions. This is a simplification of current rehearsal methods, which commonly use a separate network to generate old data, rather than storing old examples, but it still illustrates the complementary effect of utilizing both explicit replay during training and implicit spontaneous replay during sleep. The exact images from the old tasks were randomized and the fraction of old images stored was defined by degree of rehearsal. Thus, if task 1 has 5000 images and task 2 has 5000 images, then during training with 2% rehearsal, 2% of task 1 images were stored (i.e., 100 random images and their corresponding hard-target labels were stored from task 1) and incorporated into the task 2 training dataset. With more tasks (e.g., task 3), 2% of task 1 and task 2 images would be stored (i.e., 200 random images across tasks 1 and 2). In the "rehearsal" condition, we also used a weighted loss function to promote quick recovery of the old tasks as follows (see[36] for details):

$$L_{\text{total}} = \frac{1}{N_{\text{tasks}}} L_{\text{current}} + \left(1 - \frac{1}{N_{\text{tasks}}}\right) L_{\text{old}} \qquad (1)$$

When iCaRL method was tested, the fraction of stored images was defined by memory capacity, $K^{42}$, and these images were chosen based on the herding exemplar method used in iCaRL. In addition, iCaRL utilizes the nearest mean of exemplar classification scheme as well as a loss function incorporating distillation on old tasks and classification on new tasks. A more detailed description of these other methods tested in the paper can be found in the Supplementary Information and the original papers describing these methods.

## Analysis of replay

For Figs. 5 and 6 (main text), we defined the neurons that were specific to certain input classes. To define task-specific neurons, we presented inputs from each class for 25 forward passes to the network after sleep conversion and recorded the number of spikes (number of times each neuron exceeded its Heaviside-based activation threshold) in each neuron for each class of input. We ignored connections from the last hidden layer to the output layer in order to identify neurons that were more responsive to Task 1 or Task 2, while ignoring the actual classification component of the network. These spike counts were averaged across all input classes and sorted based on which input class maximally activated a given neuron. We defined task-specific neurons as the top 100 neurons that responded to a specific class of inputs. After task-specific neurons were labeled, we performed SRC and analyzed the change in activation input before and after sleep and firing rates during sleep of these task-specific neurons to create Figs. 5 and 6.

## Data availability

Data are available at the following Github repository: https://github.com/tmtadros/SleepReplayConsolidation. This includes links to the standard datasets or actual data that have been used to train the models.

## Code availability

Code is available at the following Github repository: https://github.com/tmtadros/SleepReplayConsolidation.

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

## Acknowledgements
This work was supported by the Lifelong Learning Machines program from DARPA/MTO (HR0011-18-2-0021 to M.B.), ONR (MURI N00014-16-1-2829 to M.B.), and NSF (EFRI BRAID 2223839 to M.B. and G.K.).

## Author contributions
T.T., G.K., and M.B. designed research. T.T. performed research. R.R. provided theoretical results and T.T., G.K., and M.B. wrote the paper.

## Competing interests
The authors declare no competing interests.
