## [Peer Review File · Nature Communications]

REVIEWER COMMENTS

Reviewer #2 (Remarks to the Author):

In this study, a previously developed sleep replay algorithm (SRA) is applied to the incremental training of deep neural networks (DNNs). It is demonstrated that SRA reduces catastrophic forgetting, and some insights are provided as to why SRA might be doing so.

Catastrophic forgetting is an important problem and a 'hot topic' in deep learning, and leveraging insights from the brain to address this problem has become a widely used strategy. This paper also follows this general strategy, but sets itself apart with an original and distinctive approach. The paper attempts to simulate sleep in a trained DNN, by adding an additional phase consisting of unsupervised training with local plasticity rules, with only an averaged input from the previous task as input. Interestingly, and certainly surprisingly at first, such an added sleep phase is shown to reduce the catastrophic forgetting problem on a number of common incremental learning benchmarks. The authors also provide an insightful toy experiment (FIG 1) and several interesting additional analyses (FIG 4-6) to provide insights as to why SRA might be able to reduce catastrophic forgetting.

Although the work is original and surprising, and some of the toy experiments / additional analyses provide interesting insights, I'm afraid that I cannot currently support publication of this work in Nature Communication. The main reason for this is that I am not convinced that the presented work could have practical value from a deep learning perspective (see major concern 1). Unless the authors can address this concern, this work might be better suited for a venue focussing more specifically on computational neuroscience. In addition, there is a potential issue with the toy example (see major concern 2), and it is somewhat unclear what the unique contributions of the current paper are compared to other recent work by the same group.

Major concerns

(1) My biggest concern with this paper is that, from a machine learning perspective, the achieved performance is not good. Although the authors convincingly demonstrate that their approach performs better than the naïve approach of incrementally training a deep neural network, the reported performance does not come close to that of the best performing incremental learning methods in this field, or even to a level of performance that might be indicative of value for practical applications. This relatively weak performance is probably best illustrated with the incremental MNIST problem, on which SRA obtains a performance of ~45% (or ~54% when additionally using stored data). A simple linear classifier can easily achieve >85% on this problem, and successful incremental learning methods obtain >90%. I interpret this as suggesting that SRA, at least by itself, does not provide a principled solution for the incremental learning problem.

Perhaps SRA could have practical value as a tool to enhance the performance of existing state-of-the-art incremental learning methods. In an attempt to show this, the authors combine SRA with the replay of stored data. Although the authors demonstrate that "SRA+replay" for most of their experiments performs somewhat better than "replay", the performance of the "replay" methods implemented by the authors still remains far from state-of-the-art performance. Either the amount of data that is stored is not enough, or the way that data is used is not optimal (the method section is not entirely clear about this; is the stored data simply mixed in with the training data of the new task?), or both. To demonstrate (potential) practical value, I think the authors should show that SRA could improve the performance of methods that at least approximate state-of-the-art (e.g., iCaRL).

(2) Although I think the toy example reported on in FIG 1 is insightful in several ways, I don't think I agree with the interpretation of the authors. My main concern relating to this toy experiment is that I do not understand why, during training on T1, the weights from T1-pixels to T2-output neurons do not seem to be decreased. When training on T2, the weights from T2-pixels to T1-output neurons are decreased. Why is there this difference? I think this is important, because the contribution of SRA in this toy experiment is to decrease the weights from T1-pixels to T2-output neurons, but I think this should have already happened during training on T1.

Put differently, it seems that the training processes of T1 and T2 are not symmetric (I'm not sure why they are not symmetric, as not enough experimental details are provided), and that the contribution of SRA is to correct this lack of symmetry. I'm wondering whether if the lack of symmetry between T1 and T2 is addressed directly, SRA would still have a beneficial effect.

Generalizing this observation, I wonder whether the observed benefit of SRA on the standard datasets might also be due to such a "symmetry-correcting" mechanism; and whether it is possible to correct such a lack of symmetry in the training process more directly (and without needing sleep).

Minor comments

(3) The authors should make it more clear how the current paper relates to, and how its contributions differ from, earlier work by the same group; in particular with respect to the papers Tadros et al. (2020, ICLR) and Gonzalez et al. (2020, Elife). It would be helpful to clearly state how the SRA algorithm used in the current paper differs from the sleep algorithms used in these previous papers, and it would be good to discuss in more details how the insights produced in the current paper differ from the insights that were already produced by these previous papers.

(4) It seems to me that the finding presented in FIG 6, that the activity of digit-specific neurons during sleep is greater than the activity of randomly selected neurons, could be due to the use of a "stimulating input" that is the averaged input of the previous task. Does the finding presented in FIG 6 still hold when using a random stimulating input?

(5) The description of the network architecture used for CIFAR-10 is not clear. For example, it seems from the description that the 3 VGG blocks are pre-trained before the incremental training on CIFAR-10 starts, but I could not find on what dataset (and exactly how) these VGG blocks were pre-trained.

(6) Several of the claims in the paper about SRA's ability with regards to alleviating catastrophic forgetting are too strong (e.g., in the discussion it is claimed that SRA can prevent catastrophic forgetting) and should be mediated.

Reviewer #3 (Remarks to the Author):

In this work, Tadros et al. demonstrated that introducing unsupervised learning mediated by spontaneous activity replay (SRA) can help reduce catastrophic forgetting in artificial network networks (ANNs). Improved results obtained by the proposed method were shown in a number of continual learning tasks. Possible underlying mechanisms were also explored. It was found that SRA reduced representational overlap between different classes and enhanced the representation of old tasks in continual learning. Implications to understand spontaneous activity replay during sleep in biological brains were discussed.

It is interesting that a simple Hebbian-like plasticity rule operating on top of spontaneous activity propagations in ANNs can help recover the knowledge associated with old tasks, which were apparently “erased” after learning some new, partially overlapping tasks. Better understanding its underlying mechanism would not only suggest an efficient approach to overcome the problem of catastrophic forgetting, but also provide new insights into the knowledge representation of ANNs. However, I think in both aspects, this paper still has ample room to improve.

- 1. The experiments shown in the current paper are not sufficient to demonstrate its superiority in performance.**
 - a. The comparison did not include more recently proposed regularization methods, e.g., orthogonal weight modification (Zeng et al. 2019), orthogonal gradient descent (Farajtabar et al. 2020), etc, some of which already outperformed EWC and SI significantly.**
 - b. The networks used were fairly shallow, with 2 hidden layers at most. The effectiveness needs to be demonstrated for deeper architectures.**
 - c. Based on the current description, it is not clear whether the proposed method can be successfully implemented in convolutional neural networks (CNNs). For CIFAR10 and CUB-200, the author mentioned that they used “extracted features” from CNNs. Were these CNN-based feature extractors also trained by SRA? If not, can SRA be applied to train CNNs continually?**
 - d. There are extra computational costs associated with SRA. This was not adequately analyzed in the current paper. For example, how is the performance related to the length of the “sleep” period?**

- 2. Regarding the mechanisms through which the proposed method helps overcome the catastrophic forgetting, the current analyses are mainly phenomenological, without sufficient analytical depth.**
 - a. Section B analyzed the working of SRA in a toy model. Although the phenomena revealed were interesting, no theoretical analysis was given, which casts doubt on how well this can be generalized to other conditions. The same applies to other phenomenological analyses given in the paper.**
 - b. Without a deep mechanistic understanding, SRA was implemented through a somewhat cumbersome procedure, by essentially converting ANNs to spiking neural networks (SNNs), introducing replay, modifying weights, and then converting SNNs back ANNs. To better understand what this whole procedure achieves would be very instrumental to develop a more efficient regularization method for ANNs.**
 - c. The authors reported that certain inputs are better than others in initiating the replay and obtaining its functional benefits. However, this was left unexplored. A better theoretical understanding of it would guide the search for more efficient inputs.**
 - d. It is an intriguing phenomenon that “information about old tasks remains present in the synaptic weights and can be resurrected by off-line processing”. But how is the knowledge representation stored in the configuration of synaptic weights that can sustain weights modifications by subsequent learning and be resurrected? Could this be addressed, at least in the toy system?**

Minor issues:

- 1. Sleep is a biological phenomenon and is not defined for ANNs. To use the phrase “sleep replay” in the title is somewhat ambiguous.**
- 2. It is an overstatement to say that SRA is “sufficient” to prevent catastrophic forgetting (the abstract and Line 456)**
- 3. The authors need to define “representational sparseness” and better explain how their results suggest that representational sparseness was increased by SRA. Less correlated representation (cf. Fig. 4) does not necessarily mean more sparse representation.**

4. Line 303-4: SRA “with no knowledge of the previously learned examples” is not accurate, as SRA needs to initiate reply by using inputs influenced by previously seen examples.
5. To refer to the work of Hinton et al. (1995) as “Hinton’s algorithm” is inaccurate.
6. Previous work has suggested that Hebbian learning may lead to orthogonal neural codes (cf. Saxe et al. 2020 for related discussions). How it may be related to the current results?

Ref

M Farajtabar, N Azizan, AMott, A Li. Proceedings of the Twenty Third International Conference on Artificial Intelligence and Statistics, PMLR 108:3762-3773, 2020.

A Saxe, S Nelli, C Summerfield. If deep learning is the answer, what is the question? Nature Reviews Neuroscience, 22, pages 55–67 (2021)

G Zeng, C Y, C B, S Yu. Continual learning of context-dependent processing in neural networks. Nature Machine Intelligence 1 (8), 364–372

Comments from Reviewer #2

Comment 1: *My biggest concern with this paper is that, from a machine learning perspective, the achieved performance is not good. Although the authors convincingly demonstrate that their approach performs better than the naïve approach of incrementally training a deep neural network, the reported performance does not come close to that of the best performing incremental learning methods in this field, or even to a level of performance that might be indicative of value for practical applications. This relatively weak performance is probably best illustrated with the incremental MNIST problem, on which SRA obtains a performance of ~45% (or ~54% when additionally using stored data). A simple linear classifier can easily achieve >85% on this problem, and successful incremental learning methods obtain >90%. I interpret this as suggesting that SRA, at least by itself, does not provide a principled solution for the incremental learning problem.*

Perhaps SRA could have practical value as a tool to enhance the performance of existing state-of-the-art incremental learning methods. In an attempt to show this, the authors combine SRA with the replay of stored data. Although the authors demonstrate that “SRA+replay” for most of their experiments performs somewhat better than “replay”, the performance of the “replay” methods implemented by the authors still remains far from state-of-the-art performance. Either the amount of data that is stored is not enough, or the way that data is used is not optimal (the method section is not entirely clear about this; is the stored data simply mixed in with the training data of the new task?), or both. To demonstrate (potential) practical value, I think the authors should show that SRA could improve the performance of methods that at least approximate state-of-the-art (e.g., iCaRL).

Response 1: We would like to thank the reviewer for his/her suggestion about how to better illustrate that “SRA could have practical value as a tool to enhance performance of existing state-of-the-art incremental learning methods”. Indeed, our decision to test and include “SRA+Replay” model was to show that SRA can improve performance of the other existing solutions for continual learning. In the original paper, we focused on SRA ability to significantly improve performance when old data storage capacity was very low (only 1-2% of old data was stored during new task training). With more data presented during new task training (as shown in updated Figure 3), one can see numbers start to approach the ideal accuracy, while the amount of data to be stored was still significantly less than what was required to achieve the same level of performance without SRA. We added a sentence:

“These results suggest that SRA can reduce the amount of data needed to be generated or stored with state-of-the-art rehearsal methods, while still obtaining near ideal accuracy.”

Based on your suggestion, we also implemented SRA + iCaRL model and we show that SRA does improve the performance of iCaRL alone and comes close to the ideal accuracy on the MNIST, Fashion MNIST, and CIFAR10 tasks (see new Table 2 in the paper). Here we tested cases of very small (K=100), medium (K=1000) and large (K=20000) iCaRL storage capacity. In all cases, SRA was able to exceed (K=100) or improve (K=1000, 20000) performance of iCaRL alone. In the larger capacity case, accuracy numbers approached the ideal accuracy. Table recreated here for access and described below:

	MNIST	Fashion MNIST	CIFAR 10
SRA	44.87±0.0630	42.03±0.0175	34.56 ± 0.012
iCaRL, K = 100	19.178 ± 0.631	22.652 ± 3.714	18.444 ± 0.626
iCaRL, K = 1000	56.528 ± 2.921	45.376 ± 2.268	33.858 ± 2.283
iCaRL + SRA, K = 1000	73.154 ± 5.922	56.600 ± 5.482	49.206 ± 3.837
iCaRL, K = 20000	93.000 ± 0.393	80.280 ± 1.092	66.156 ± 0.924
iCaRL + SRA, K = 20000	94.822 ± 0.235	83.220 ± 0.735	66.304 ± 0.869
Ideal ANN	94.19 ± 0.001	83.48 ± 0.003	67.36 ± 0.002

The first two rows show a comparison of SRA vs. iCaRL with K=100. SRA alone has equivalent memory capacity of K=1 since it stores only average input. We found that SRA alone showed significantly higher performance compare to iCaRL with K=100. Moving to K=1000, SRA was able to substantially improve the performance of iCaRL, by 10-40% on the different datasets, when integrated together. Finally, moving to K=20000, both iCaRL alone and iCaRL+SRA approached the accuracy of an ideal network, with iCaRL+SRA sometimes exceeding this (in the case of MNIST). This suggests that SRA can be complementary to rehearsal methods and can significantly reduce the amount of data needed to achieve state-of-the-art performance. A new section (section D in the Results section) has been included to the revised manuscript to present these results.

We would also like to make a more general comment. We certainly agree that any new machine learning approach should be first and primarily evaluated based on its actual performance. As we show above, SRA can be easily combined with almost any existing machine learning (ML) model and it can be very useful as a complimentary approach to existing machine learning solutions to reduce data storage but its performance alone, at least in its current implementation, is still below that of some other state-of-the-art methods. Saying that, we would like to suggest that another “value” of a new approach is to highlight new directions that can be further explored and improved in future studies. Human brain does not store any old data and sleep along with unsupervised sleep replay play central role in learning in biology. We do not know if sleep replay (in biological sense, with no data stored) can be an ideal solution to prevent catastrophic forgetting in exiting ANN models (brain does not use backpropagation to learn in the first place) but at least our study suggests that some principles from neuroscience can be applied to improve existing ML algorithms. Brain uses many complimentary plasticity mechanisms to optimize memory storage during sleep and our current solution, while capturing few important mechanisms, is certainly a huge simplification of reality, which may be another reason why SRA performance alone is still suboptimal.

Comment 2: *Although I think the toy example reported on in FIG 1 is insightful in several ways, I don't think I agree with the interpretation of the authors. My main concern relating to this toy experiment is that I do not understand why, during training on T1, the weights from T1-pixels to T2-output neurons do not seem to be decreased. When training on T2, the weights from T2-pixels to T1-output neurons are decreased. Why is there this difference? I think this is important, because the contribution of SRA in this toy experiment is to decrease the weights from T1-pixels to T2-output neurons, but I think this should have already happened during training on T1. Put differently, it seems that the training processes of T1 and T2 are not symmetric (I'm not sure why they are not symmetric, as not enough experimental details are provided), and that the contribution of SRA is to correct this lack of symmetry. I'm wondering whether if the lack of symmetry between T1 and T2 is addressed directly, SRA would still have a beneficial effect.*

Generalizing this observation, I wonder whether the observed benefit of SRA on the standard datasets might also be due to such a “symmetry-correcting” mechanism; and whether it is possible to correct such a lack of symmetry in the training process more directly (and without needing sleep).

Response 2: This is great question. Let us first try to address the “specific” part of the question about why during T1 training do weights from T1-pixels to T2-output neurons not become negative. In the original paper, we were using a ReLU activation function at the output layer. Thus, there is no difference between an output of -10 and -0.1 when passed to the ReLU function. Hence, the weights were only modified enough to cause 0 activity in the T2-output

neurons during T1 training. We have since updated the plot to use the Softmax activation function, and here we can see that weights from T1-pixels to T2-output neurons are decreased during T1 training (new figure 1 and related text in the revised paper).

Using softmax has not prevented catastrophic forgetting. We have added a new panel (Figure 1C) to explain this. This figure shows the sum of the weights from all T1 pixels (e.g. T1 unique pixels and T1/T2 overlapping pixels) to T1 output neurons in red, and the sum of the weights from all T1 pixels to T2-output neurons in black, after Training T1, Training T2, and SRA. If the black bar is greater than the red bar, the network will predict a T1 class. If the red bar is greater than the black bar, the network will (incorrectly) predict a T2 class. Thus, you can see that after training on Task 2, the network is mistaking T1 images for T2 classes as T1 images preferentially activate T2 output neurons. This is reversed after SRA. The new figure is included to the revised version and presented below for clarity:

Figure 1: SRA reduces catastrophic forgetting for sequential task training. A) Example of four binary images with 15 pixel overlap divided into task 1 (T1, images 0,1) and task 2 (T2, images 2,3). T2 was trained after T1. B) Classification accuracy as a function of number of overlapping pixels (interference) after training T1 (blue), T2 (red), and SRA (yellow). The network was always tested for all 4 images. Note, that performance is significantly higher after SRA than before in the range of pixel overlap 10-20. C) Input to the output neuron 0 when presented with image 0 (black) and the max input to the output neurons 2 and 3 when presented with image 0 (red) after subsequent stages of training and sleep. D) Distribution of all weights connecting to T1 (left) or T2 (right) output neurons. Color is based on category of connections: Unique - connections from neurons representing pixels unique for T1 or T2; Overlapping - connections from neurons representing pixels that overlap between T1 and T2. Rows show subsequent stages of training and sleep

Now, responding to the “general” part of the question, indeed, it makes great sense that some form of “symmetry-correcting” mechanism can prevent catastrophic forgetting by making each task’s representation in the network to be roughly equal (in terms of activation). And this is what we believe, based on the neuroscience data, sleep uniquely does by co-replaying old and new memories, so each of them would reach a new optimal representation that allows coexistence with competing memories. The question is how to achieve this goal in ANNs. For example, regularization methods aim to preserve weights that were important

for old tasks while modifying weights that are less important to represent the new tasks.

However, these methods performance is sub-state-of-the-art on class-incremental learning tasks or require complex calculations to ensure orthogonality of representations.

From that perspective sleep algorithm presented here indeed may be viewed as a symmetry-correcting mechanism. As reviewer #3 points out, Hebbian learning has been shown to orthogonalize neural codes and our biophysical modelling work (Gonzalez, et al., eLife 2020) has shown that sleep can indeed orthogonalize memories by assigning distinct weights to represent distinct memories. In that study, we showed (using biophysical models that are absolutely not practical from ML point of view) that rather than engage completion between memories, sleep allows chunks of competing memories to replay simultaneously/together, so each memory would reach its optimal representation. So, indeed, it recovers symmetry! There may be other “symmetry-preserving” methods that achieve this goal. However, we believe that the biological-realism presented in SRA can further add to this literature by showing that Hebbian learning rules can reverse the imbalance in the network obtained after any new task training to support recollection of previous memories. We included a short discussion of this issue in the discussion section to stimulate further research on this important question:

“Catastrophic forgetting may be interpreted as asymmetry of the weights’ configuration that became biased towards the most recent task after new training. From that perspective, SRA may be seen as a “symmetry-correcting” mechanism. Indeed, Hebbian learning has been shown to orthogonalize neural codes and our recent biophysical modelling work (Gonzalez, et al., eLife 2020) has shown that sleep orthogonalizes memories by “assigning” distinct weights to represent distinct memories. We found that rather than engage completion between memories, sleep allows chunks of competing memories to replay simultaneously, so each memory would reach its optimal representation, which can be seen as a way of recovering the symmetry of weight distribution across tasks.”

Minor Comments

Comment 3: *The authors should make it more clear how the current paper relates to, and how its contributions differ from, earlier work by the same group; in particular with respect to the papers Tadros et al. (2020, ICLR) and Gonzalez et al. (2020, Elife). It would be helpful to clearly state how the SRA algorithm used in the current paper differs from the sleep algorithms used in these previous papers, and it would be good to discuss in more details how the insights produced in the current paper differ from the insights that were already produced by these previous papers.*

Response 3: Let us first describe how the current paper differs from the ICLR paper. In the ICLR paper, we were addressing the problem of increasing robustness to noise and adversarial attacks; the problem of continual learning was not studied at all. For a brief summary: when generic ANN is trained on normal, intact images, but tested on noisy (blurry, gaussian noise, etc.) images, its performance will suffer. We proposed that subsequent sleep replay could

provide a mechanism to mitigate this problem, as it has been hypothesized in neuroscience research that sleep can help create a generalized representation of memories so that they become more robust (stable and predictable). Since this problem does not have explicit memory capacity constraints, as there are for continual learning task (where ideally a model do not have access to old tasks during new task training), the sleep algorithm in ICLR paper was implemented using inputs from actual training datasets. In our new study, we changed the algorithm to explore the problem of continual learning by taking into account the memory constraint and we only stored an average input to use during sleep replay. It may be, however, important to notice that conceptually similar approaches, both utilizing unsupervised sleep replay during post training processing, may help for two very different ML problems (adversarial robustness and continual learning), suggesting potentially generic role of sleep principles explored in our studies.

As for Gonzalez et al., our new paper is quite different by tackling a different setting (ANNs, image classification), whereas Gonzalez et al., looked at representations of very simple overlapping memory sequences (e.g, sequential activation of neurons $A \rightarrow B \rightarrow C$, $C \rightarrow B \rightarrow A$) in a biophysical model of the thalamocortical system. Because of realism and complexity of Hodgkin-Huxley neurons and thalamocortical model design, eLife study allows to make a strong connection to experimental data but “hopeless” from ML perspective as proposed biophysical model cannot learn anything more complex than few letters sequence. However, our findings are complementary: both suggest that sleep can recover symmetry of the memory representations (see discussion above) by devoting certain neurons to one task and other neurons to another task. Our new findings here are also conceptually novel compared to eLife paper: Hebbian learning can be applied in other domains with beneficial results; ANNs may maintain information about old tasks even when not apparent from performance-level analysis. In addition, we also make new testable predictions for neuroscience, such as sleep increases the sparseness of memory representations in the network.

We have added the following statements to the paper:

1) 2nd paragraph in discussion:

“While actual training data were used for sleep replay in that study, here we show that sleep replay can alleviate catastrophic forgetting on class incremental learning by only having access to the basic input statistics.”

2) 1st paragraph in Section E:

“Using biophysical models of brain network and testing for simplified task of learning overlapping memory sequences, we showed that sleep replay modifies the synaptic weights to create unique synaptic representation for each task [24]. “

Comment 4: *It seems to me that the finding presented in FIG 6, that the activity of digit-specific neurons during sleep is greater than the activity of randomly selected neurons, could be due to*

the use of a “stimulating input” that is the averaged input of the previous task. Does the finding presented in FIG 6 still hold when using a random stimulating input?

Response 4: We would like to thank the reviewer for this suggestion. We have added a supplemental figure showing that our finding remains even if we use a random input for network stimulation during sleep.

Supplemental Figure 3: Spiking activity of digit-specific neurons during sleep is greater than activity of random subset of neurons in both first (top row) and second (bottom row) hidden layers. Here, a random input is presented during sleep, rather than an input with statistics based on the training set.

Also, this sentence was included:

“Importantly, this result holds even when completely random inputs during sleep are used instead of inputs based on average image statistics (Supplemental Figure 3).”

Comment 5: The description of the network architecture used for CIFAR-10 is not clear. For example, it seems from the description that the 3 VGG blocks are pre-trained before the incremental training on CIFAR-10 starts, but I could not find on what dataset (and exactly how) these VGG blocks were pre-trained.

Response 5: Thanks for pointing to this place of confusion. We pre-trained the 3 VGG blocks on the tiny imagenet dataset. After pre-training, the weights were frozen and the intermediate-level representations were computed for all classes in the CIFAR-10 dataset, in order to run the incremental-learning paradigm. We have added specific information (learning rate, etc.) about this training procedure in the methods section:

“The original convolutional backbone was trained on the Tiny Imagenet dataset and CIFAR-10 images were fed through this network to extract intermediate feature representations \cite{le2015tiny}. The network was trained for 200 epochs using stochastic gradient descent with the following parameters: momentum = 0.9, learning rate = 0.005, batch size = 100, and the categorical crossentropy loss function.”

Comment 6: *Several of the claims in the paper about SRA's ability with regards to alleviating catastrophic forgetting are too strong (e.g., in the discussion it is claimed that SRA can prevent catastrophic forgetting) and should be mediated.*

Response 6: Thanks for bringing this up - we have modified a few of the claims made in the paper. See response to Reviewer #3, comment 4, for specific changes made in the paper.

Response to Reviewer #3

Comment 1: *The experiments shown in the current paper are not sufficient to demonstrate its superiority in performance.*

Response 1: We agree that results presented in the original manuscript were not sufficient to highlight SRA's performance. We have addressed this concern in two ways:

- 1) We included additional analysis and comparison to iCaRL to further bolster the claim that SRA can be complementary to other rehearsal methods by reducing the amount of data that needs to be stored or generated. The new results demonstrate that SRA can be combined with other methods to approach the upper bound of an ideal learner with less data than rehearsal methods - please see our detailed response to the Comment 1 by Reviewer #2, including table showing performance of SRA+iCaRL. These results are summarized in the new section D of the revised manuscript.
- 2) We performed additional experiments as suggested by the reviewer to compare SRA with orthogonal weight modification. (Please see below our response to specific comments)

Before describing additional new results of comparing SRA to regularization methods, we would also like to point out the potential value of our work from a standpoint of developing brain inspired AI algorithms. From an algorithmic standpoint, advances in accuracy, feasibility, and computational time over existing approaches are the most important criteria for evaluating any learning system. While current standalone SRA implementation is still below that of some other state-of-the-art methods, SRA can be easily combined with almost any existing machine learning (ML) model and it can be very useful as a complimentary approach to existing machine learning solutions to reach an ideal learner performance still using reduced data storage (please see our response to Reviewer 2, Comment 1). However, we believe that our work stands out for not only its accuracy and efficiency, but also because it highlights new neuroscience inspired directions that can be further explored in future studies. This work suggests that learning principles found in the brain - local unsupervised synaptic plasticity occurring during dedicated off-line (sleep) phase - can be applied to the current ANNs to mitigate catastrophic forgetting. Although its accuracy as a standalone algorithm may not be as impressive compared to state-of-the-art methods, it is still a compelling story that an unsupervised learning rule inspired from biology can have such an effect and achieve similar/greater results than some other supervised methods, such as the regularization methods discussed below. This type of work can make predictions in

neuroscience as pointed out in the discussion and below, thus not only serving as valuable to the computer science community but also to the neuroscience community.

Comment 1a: *The comparison did not include more recently proposed regularization methods, e.g., orthogonal weight modification (Zeng et al. 2019), orthogonal gradient descent (Farajtabar et al. 2020), etc, some of which already outperformed EWC and SI significantly.*

Response 1a: Thank you for these references. We tested our tasks using *orthogonal weight modification* (Zeng et al. 2019) and have added these results to Table 1 in the main paper. There was no code repository for *orthogonal gradient descent*, so we have left this study out. As the reviewer suspected, OWM does indeed surpass EWC and SI on class-incremental tasks. OWM does show some performance advantage over SRA on most tasks. However, both methods are relatively comparable and are not reaching ideal accuracy when implemented alone. We included the following text:

“However, a recently developed regularization method (Orthogonal Weight Modification, OWM) slightly surpasses performance of SRA on most tasks, suggesting that regularization methods can promote recovery of old tasks in a class-incremental learning setting. \cite{zeng2019continual}. One implementation advantage of SRA over OWM is that SRA is an offline method, so it can be run after the normal training process is completed. Therefore, it can be complimentary to any other continual learning method. Furthermore, for tasks where it is unknown a priori if/when new training would be needed, SRA can be easily applied post fact, as only average input from data used for preceding training would be needed to run SRA. In contrast, with OWM, one will need to feed all the previous inputs through the network to compute the projections.”

As noted in our response to Reviewer #2 (Comment 1), we also did a more systematic test of SRA + rehearsal and were able to observe near-ideal performance on incremental learning tasks while using significantly less data than rehearsal methods alone (new Table 2). The authors of the OWM method have also noted that their method may be complementary to replay (Shen et al., 2021). We leave it to future work to explore which method requires less data to be replayed and thus would be more data-efficient.

Comment 1b: *The networks used were fairly shallow, with 2 hidden layers at most. The effectiveness needs to be demonstrated for deeper architectures.*

Response 1b:

As noted in the response to Comment 1c below, we believe SRA only needs to be applied to the top-most layers of the network. However, to address the reviewer concern, we applied SRA to 4 hidden layer networks on the class-incremental MNIST task in two settings: (a) the first 2 layers are frozen after first task training and only last two are changing during subsequent training or SRA; (b) all 4 layers are plastic during each task training and SRA. With 2 hidden layers

network (main configuration we tested in this paper), the ANN+SRA method is able to achieve an entire dataset classification accuracy of 44.87%, while the naive network (ANN alone) achieves classification accuracy of 19.26%. With 4 hidden layers network (all layers are the same size, with 1200 units) and all 4 layers being plastic, the ANN+SRA method can achieve a classification accuracy of 46.04% whereas a naive network achieves 19.34% classification accuracy. If the first 2 hidden layers are frozen after first task training (so only the last 2 hidden layers are plastic), then ANN+SRA achieves 42% classification accuracy whereas ANN alone achieves 18.6% classification accuracy. This shows that performance (at least, in the case of MNIST) does not depend significantly on the number of layers, and with more layers, SRA only needs to be applied to the top-most layers to alleviate catastrophic forgetting (please also see our response to Comment 1c below). We added a sentence to the text to briefly mention the result with 4 layers network:

“While we primarily tested a network with only two hidden, the analysis of 4-hidden layer networks on MNIST task revealed that SRA can be applied in deeper networks to recover the same level of performance.”

Comment 1c: *Based on the current description, it is not clear whether the proposed method can be successfully implemented in convolutional neural networks (CNNs). For CIFAR10 and CUB-200, the author mentioned that they used “extracted features” from CNNs. Were these CNN-based feature extractors also trained by SRA? If not, can SRA be applied to train CNNs continually?*

Response 1c: Thank you for this important comment. We have not yet tried to implement SRA for CNNs. In continual learning settings (especially in class-incremental learning), it may not be necessary to tune the CNN backbone of the architecture to prevent catastrophic forgetting. Indeed, it was demonstrated in several recent studies (Zheng et al. 2019, Kemker et al. 2018), that using a pre-trained feature extractor and building a continual learning-enabled network on top of the extractor is sufficient to prevent catastrophic forgetting. We leave open the possibility of incorporating SRA into the training phase of an end-to-end CNN architecture. However, in the current domain tested in the paper, it was not necessary to modify CNN weights to prevent catastrophic forgetting. In fact, the literature has suggested that catastrophic forgetting does not occur in the feature extractor but occurs in the later layers, which motivated the use of SRA in these layers (Goodfellow et al., 2013; Ramasesh et al., 2020).

This idea is similar to fine-tuning, in which the convolutional backbone is fixed during fine-tuning and only the later layers are modified to adapt to the new classification paradigm. Since the CNN computes generalized features such as basic image statistics and properties, it is unlikely that catastrophic forgetting would occur in these layers. From neuroscience perspective convolutional layers are somewhat equivalent to the primary visual cortex that is less plastic

beyond development and unlikely involved in significant rewiring during sleep replay, in contrast, e.g., to the associative cortices. We included a following statement into the Discussion:

“It remains open question if SRA would benefit convolution layers of the network, which were kept frozen in this study. Recent studies suggest that catastrophic forgetting does not occur in the feature extractor but takes place in the later layers, which motivated the use of SRA in these layers (Goodfellow et al., 2013; Ramasesh et al., 2020). From neuroscience perspective convolutional layers are somewhat equivalent to the primary visual cortex that is less plastic beyond development and unlikely involved in significant rewiring during sleep replay, in contrast, e.g., to the associative cortices.”

Comment 1d: *There are extra computational costs associated with SRA. This was not adequately analyzed in the current paper. For example, how is the performance related to the length of the “sleep” period?*

Response 1d: We first note that length of sleep is mostly related to the size of the network, not to the size of the training set. Thus, for larger datasets, while more images may be needed during training phase (in the “awake” state) in order for classification accuracy/loss to saturate, during “sleep” this is not necessarily true. Because of this, the computational cost of sleep is generally similar for networks trained on Imagenet, CIFAR, MNIST, etc (as long as the network size that sleep is applied to is the same, e.g., 2 hidden layers with the same number of units).

As for number of iterations of sleep, we have observed that two parameters dictate how much sleep is necessary to recover performance on old tasks: the magnitude of weight changes (determines how much to increase/decrease weights when an STDP event occurs) and the input firing rate (that is the maximum firing rate for neurons in the first layer during sleep). Parameters can be set so that a very small amount of sleep is needed to recover old task performance. In the following figure, included as Fig 2 to the Supplementary Materials, one can see that as the network learns all 5 MNIST tasks, only ~500 iterations (and thus ~500 feedforward passes through the network) of sleep are needed to reach performance saturation. Each line shows the accuracy as a function of sleep duration on the entire dataset (all 5 tasks). Color indicates the sleep phase (e.g., after training task 1, task 2. etc.)

Supplemental Figure 2: Accuracy as a function of sleep iteration for MNIST (left) and CIFAR10 (right) datasets. Each line represents the application of SRA after a specific training phase (e.g., after training T1, T2, etc).

If sleep requires 500 iterations to ultimately converge, this is equivalent to presenting 500 “noisy” images. In contrast, during training, if we train each class for 2 epochs (as in the case with MNIST), 10,000 images are presented twice, i.e., 20,000 images fed through the network both forwards and backwards for a total of 40,000 passes. SGD is efficient so we do this in batches and we should divide the total number of passes (feedforward + backward) by the batchsize (100). This means the training computational time is $40,000/100 = 400$ passes. This is on the same level as what is required during sleep (400-500 passes till convergence). The computational performance of sleep phase can be likely further improved (e.g., by incorporating the idea of mini-batches, etc.), just as how gradient descent has been heavily optimized.

Lastly, in terms of memory constraints, there are scenarios where SRA reduces storage requirements (as shown with iCaRL) and scenarios where SRA requires more memory (as compared to regularization methods). However, regularization methods may be more computationally intensive, as they require computation of complex matrices, e.g. Fisher information matrix in EWC, or orthogonal projectors in OWM.

We included the text and fig above in Supporting Information and included the following sentences to the revised text:

“Note that computational costs for running SRA are comparable with the costs for training each additional task (when task training is implemented in batches). However, sleep required much less inputs to pass through the network; thus, the computational performance of sleep phase can be likely improved by incorporating the idea of mini-batches (see Supporting Information for more detail).”

Comment 2: *Regarding the mechanisms through which the proposed method helps overcome the catastrophic forgetting, the current analyses are mainly phenomenological, without sufficient analytical depth.*

Response 2: Reviewer is correct. We would like first to say that it also remains a major question in neuroscience, to characterize rigorously how sleep can give rise to a plethora of different memory benefits. Although this revision does not present a satisfactory analytical explanation of the benefits of SRA, we would like to point out the benefits of the phenomenological analyses presented here in guiding future neuroscience research. First, we show that during sleep, representation of distinct tasks becomes decorrelated in the network, with different sets of neurons devoted to representing different tasks. This idea gives rise to the hypothesis that during sleep, overlapping items representations may become decorrelated and could spur experimentation of this hypothesis. In addition, we show that sleep can balance activity between two tasks, mitigating any recency bias in learning and that prolonging new task training can actually reduce the benefit of sleep (see below). These predictions also support a testable hypothesis that can guide further research in the mammalian brain. While we would like to leave it to future work to come up with a complete analytical explanation of the benefits of sleep in generic ANNs, we provided additional analysis of sleep replay in toy model (see below).

Comment 2a: *Section B analyzed the working of SRA in a toy model. Although the phenomena revealed were interesting, no theoretical analysis was given, which casts doubt on how well this can be generalized to other conditions. The same applies to other phenomenological analyses given in the paper.*

Response 2a:

We thank the reviewer for pointing out to this issue. We now included analysis that provides some theoretical understanding of how catastrophic forgetting occurs and how sleep can mitigate it in toy model. Thus, we show that in case of a perceptron, as well as multilayer perceptron (MLP), trained for two tasks sequentially, the network parameters following training of a new class have positive cosine similarity to the parameters of the network trained on the old class alone (See new section in Supporting information). This demonstrates that even when catastrophic forgetting is observed from classification perspective, the network weights preserve information about previous tasks. This, along with the finding that the sleep phase promotes reactivation of the previous tasks (Fig 6 and supplementary fig 3,7), suggest that a local unsupervised plasticity during spontaneous network activation can change the network parameters towards the old task's configuration.

The following text was included to the main paper:

“A more rigorous analysis of this toy model revealed that following training of T1-T2 the network weights have positive cosine similarity to the weights of the network trained on T1 alone (See Supporting Information, section 1). This demonstrates that even when catastrophic

forgetting is observed from classification perspective, the network weights preserve information about previous tasks. Further, this prediction can be generalized to the multilayer perceptron.”

Comment 2b: *Without a deep mechanistic understanding, SRA was implemented through a somewhat cumbersome procedure, by essentially converting ANNs to spiking neural networks (SNNs), introducing replay, modifying weights, and then converting SNNs back ANNs. To better understand what this whole procedure achieves would be very instrumental to develop a more efficient regularization method for ANNs.*

Response 2b:

This is another great suggestion also related to the previous comments about analytical understanding the effects of SRA. We attempted to address this question in the analysis presented in the section E. Specifically, in fig.4 we demonstrate that SRA reduces correlations (in a sense of activations) between image classes but maintains it within classes; in fig.4 we show that SRA is trying to balance the network by reducing input to the output neurons representing most recent class; in fig. 5 we show that SRA mostly activates nodes that represent previously learned tasks, i.e., effectively execute replay of all the trained memories. While in this revised version we extended some of these analysis (see Supplemental Figure 3, where network is driven by pure noise), further exploration may be needed to derive rigorous regularization approaches approximating SRA functions.

While we believe some progress may and will be done in this direction, complexity of the problem may be related to the complexity of the sleep function in the biological brain (Oudiette et al., 2013 J. Neuro; Paller & Voss, 2004 Learn. & Mem.; Rasch & Born, 2013 Psychological Reviews; Stickgold, 2013 Curr Opin Neurobiol; Walker & Stickgold, 2004 Neuron). Role of sleep in application to memory is extremely multifunctional and is not limited by reducing memory interference. Indeed, the very basic function of sleep is to improve memories (memory consolidation), i.e., memory performance is improved after sleep (at least for procedural memory tasks, such as learning new skills, e.g., playing tennis). Sleep is further involved in reducing interference between similar memories, generalization of memories, transfer of knowledge, etc. Surprisingly, we found that some of these very different functions can be achieved by conceptually similar replay algorithms. In this current paper we focus on the effect of sleep replay to reduce interference between memories; in a related study (Tadros et al., 2020 ICLR) we showed effect of SRA for improving generalization and adversarial robustness. Possibly, the most basic effect of SRA, which may also help to understand its mechanistic effect, is demonstrated by the new analysis we included here. In new Fig. 6 in Supporting Information, we show effect of SRA on a single task performance as a function of the amount of initial training. What we observe is that when the network is undertrained, i.e., initial performance is low, SRA can greatly increase it without involving any new training data. However, if memory is well trained, SRA cannot improve or even slightly reduces performance. This is consistent with

neuroscience data suggesting that replay - reactivation during sleep of the neuronal patterns representing recently learned memories - can lead to performance improvements only when the network still “have a room” to modify synapses to further improve memory recall. Indeed, recent study revealed an inverse association between learning performance and gains in procedural skill in humans, i.e., good learners exhibited smaller performance gains after sleep than poor learners (Raangtell et al., 2017 Scientific Reports). However, even this, possibly the simplest effect of sleep on a single task performance, cannot be easily formalized – simply scaling up the strongest weights in the undertrained network does not lead to performance improvements (Fig. 6 in Supplement). Saying this, we realize that these analyses do not directly addresses reviewer question. To some extent the point we are trying to make here is to bring attention of ML community to the underexplored role of sleep, to stimulate further analysis that hopefully could lead to more rigorous regularization algorithms.

Supplemental Figure 6: Classification accuracy on the MNIST dataset before (black) and after (red) sleep when trained for 10, 20, or 40, 150 epochs. Scaling of top 2.5% of weights by 1% of trained value shown in grey.

The following text was included:

“If replay indeed provides a mechanism of how sleep protects memories from interference, then for a single task, sleep replay should increase performance. Indeed, the most well-established effect in neuroscience of sleep on memory is either mapping memory traces from hippocampus to the cortex for declarative memory tasks or augmenting cortical traces for procedural memory tasks \cite{rasch2013sleep}. In this new study we typically observed some reduction in performance on most recent task. Do we have contradiction here? To test this, we tested effect of SRA on a single task memory performance as a function of the amount of initial training. We found (Supplementary fig. 6) is that when the network is undertrained, i.e., initial performance is low, SRA can greatly increase performance without involving any new training data. However, if

memory is well-trained, SRA cannot improve or even slightly reduces performance. This is consistent with neuroscience data suggesting an inverse association between learning performance pre-sleep and gains in procedural skills post-sleep in humans, i.e., good learners exhibited smaller performance gains after sleep than poor learners (Raangtell et al., 2017 Scientific Reports).”

Comment 2c: *The authors reported that certain inputs are better than others in initiating the reply and obtaining its functional benefits. However, this was left unexplored. A better theoretical understanding of it would guide the search for more efficient inputs.*

Response 2c: Thanks for this suggestion. We have conducted an analysis of the effect of input type during SRA on the MNIST task performance. Three types of inputs were compared: 1) random noise, 2) average noise (i.e., noise with statistic that matches properties of the training data), 3) specific images (from training dataset). Each input was presented either using the entire image (i.e., the entire 28x28 image is used to define a Poisson distributed input during each iteration of sleep) or as a mask (a randomly selected in each sleep iteration 10x10 square of the input image is selected to define Poisson distributed input during each iteration of sleep). We have included new results to the Supporting Information. Not surprisingly, using specific inputs during sleep performed the best. However, using the average input was able to achieve ~90% of the classification accuracy of the specific inputs. Using random noise performs much worse, but this may be due to some extend to the dataset, where most of the structure of the MNIST images is in the center of the image and having activity on the borders would lead to little/no activity in the network as these weights will be near-zero or negative.

Supplemental Figure 4: Classification accuracy on class-incremental MNIST task with different types of inputs (average, random, specific) with 2 conditions (mask/no mask).

Comment 2d: *It is an intriguing phenomenon that “information about old tasks remains present in the synaptic weights and can be resurrected by off-line processing”. But how is the knowledge representation stored in the configuration of synaptic weights that can sustain weights modifications by subsequent learning and be resurrected? Could this be addressed, at least in the toy system?*

Response 2d: We included a new analysis (Figure 1C) about how weights configuration still preserve information about old tasks. In Figure 1C we show the sum of the weights from T1 pixels (e.g. T1 unique pixels and overlapping pixels) to T1 output neurons in red, and the sum of the weights from T1 pixels to T2-output neurons in black, after Training T1, Training T2, and SRA. After training T1, the black bar is significantly positive and the red bar is significantly negative. After training T2, the red bar becomes positive, but the black bar does not become negative, illustrating that T1 images will still activate T1 output neurons, but not as strongly as they activate T2 output neurons, leading to misclassification. After SRA, the black bar remains positive and the red bar became negative again, i.e., T1 inputs are correctly classified. We added a sentence to the main text to emphasize this point:

“It is worth mentioning that after training T2, the input to correct T1 output neuron was reduced but did not become negative (Fig. 1C, black bar in middle group). Thus, T1 images would still activate T1 output neurons, but not as strongly as they activate T2 output neurons -- misclassification but not complete forgetting of the T1 weight structure, so it still can be resurrected by SRA.”

From more theoretical perspective, we included a proof that training new task in toy model preserves information about synaptic weights representing the old task in perceptron and multilayer perceptron cases (see our response to Comment 2a above).

Outside of this toy example, we have also done an analysis of how length of task 2 training affects recovery after sleep. If we increase the length of task 2 training, one can imagine that the sum of the weights contributing to activation of correct output neurons for the old tasks may become negative and then recovery after sleep would be more difficult or impossible. This hypothesis was confirmed, as increasing the length of task 2 training does result in less recovery of task 1 performance, as shown in the following figure (note that Pre-sleep T1 accuracy is zero, so dashed red line overlaps with X-axis):

Supplemental Figure 5: Classification accuracy as a function of length of task 2 training (in epochs) for 2 task MNIST, Fashion MNIST, and CIFAR 10 as well as the Cross Modal MNIST task. Solid red - T1 accuracy after sleep, dashed red - T1 accuracy before sleep (after T2 training), dashed gray - T2 accuracy before sleep (after T2 training), and solid gray - T2 accuracy after sleep.

This indeed suggests that increasing training on a new task may “erase” existing old task representations from the network.

Minor Comments

Comment 3: *Sleep is a biological phenomenon and is not defined for ANNs. To use the phrase “sleep replay” in the title is somewhat ambiguous.*

Response 3:

We replaced it by “Sleep-like Unsupervised Replay Reduces Catastrophic Forgetting in Artificial Neural Networks”

Comment 4: It is an overstatement to say that SRA is “sufficient” to prevent catastrophic forgetting (the abstract and Line 456)

Response 4: We have replaced it with “SRA can alleviate” or similar in all such cases.

- 1) We have modified the last sentence of the abstract (a) to the new sentence (b):
 - a) “The study suggests that local, biologically inspired learning rules applied during spontaneously generated replay are sufficient to alleviate catastrophic forgetting in ANNs.”
 - b) “The study suggests that local, biologically inspired learning rules applied during spontaneously generated replay can alleviate catastrophic forgetting in ANNs.”
- 2) Line 456 was originally (a) and is now (b)

- a) “Our results suggest that unsupervised Hebbian type plasticity combined with spontaneous activity during sleep can help alleviate catastrophic forgetting in an incremental learning setting.”
- b) “Our results suggest that unsupervised Hebbian type plasticity combined with spontaneous activity during sleep can help alleviate catastrophic forgetting in an incremental learning setting.”

Comment 5: *The authors need to define “representational sparseness” and better explain how their results suggest that representational sparseness was increased by SRA. Less correlated representation (cf. Fig. 4) does not necessarily mean more sparse representation.*

Response 5: We have added a new supplemental figure showing that sleep does indeed result in more sparse representation. Representational sparseness is defined as inverse to a fraction of neurons that are strongly activated by stimulus. Thus, representational sparseness increases when a stimulus is represented by a strong activation of a smaller subset of neurons. We show that representational sparseness increases following SRA by demonstrating that the number of neurons responding to a digit class is reduced after SRA (Supplemental Figure 8). The following text has been added to the main paper:

“After SRA we observed decorrelation (near-zero correlations between representation of digits from different classes) and an increase in representational sparseness (where each stimulus strongly activates only a small subset of neurons) for all four digits (Figure 4, right and Supplemental Figure 8).”

Supplemental Figure 8: Average activation of 40 randomly selected neurons before (left) and after sleep (right) when presented with digits from each class. Overall, less neurons respond to each digit suggesting a sparser representation

Comment 6: *Line 303-4: SRA “with no knowledge of the previously learned examples” is not accurate, as SRA needs to initiate reply by using inputs influenced by previously seen examples.*

Response 6: Thanks, this has been modified. The new statement claims:

“SRA was able to reduce catastrophic forgetting with only limited (average statistics) knowledge of the previously learned...”

Comment 7: *To refer to the work of Hinton et al. (1995) as “Hinton’s algorithm” is inaccurate.*

Response 7: This has been fixed, thanks for the correction. It is now referred to as “The wake-sleep algorithm” and is cited properly (*Hinton et al., 1995*).

Comment 8: *Previous work has suggested that Hebbian learning may lead to orthogonal neural codes (cf. Saxe et al. 2020 for related discussions). How it may be related to the current results?*

Response 8: Thank you for this reference, this is a very interesting point. This paper (*Saxe et al. 2020*) points to a resource suggesting that Hebbian learning may lead to orthogonal neural codes for items with different temporal contexts (*Bouchacourt et al., 2020*). For example, a stimulus in one context may lead to one response, but that stimulus in another context may lead to a different response. Our work may further add to this by stating that for different memories with similar contexts (or where there is no context to distinguish items, as is the case with class-incremental image classification), Hebbian plasticity may lead to developing orthogonal neural codes. These results may be further related to our recent study of sleep in biophysical thalamocortical model of deep sleep with STDP (*Gonzalez, et al., eLife 2020*), where we report that sleep replay leads to orthogonalization of the neuronal representations for memories representing competing neuronal sequences.

The following sentences have been added in reference to these two articles:

“Recent studies suggest that local learning rules can orthogonalize memories with different temporal contexts (*Bouchacourt et al., 2020; Saxe et al., 2020*). Our current (and recent (*Gonzalez, et al., eLife 2020*)) work agrees with this idea and further suggests that local learning rules and sleep can orthogonalize representation of interfering memories, even with limited contextual information.”

REVIEWER COMMENTS

Reviewer #2 (Remarks to the Author):

Thanks to the authors for the updated manuscript and an extensive response to my comments.

Despite the addition of several new experiments, my main concern with this paper is still that, from a practical machine learning perspective, the value of the proposed SRA is not convincingly demonstrated. Therefore, I cannot support acceptance of this paper.

One important reason for this is that several of the reported experiments, especially some of the newly added ones, seem not to have not been performed to the expected standards. - As indicated in my original review, I had doubts about the level of performance that the authors obtained using "replay". I had asked the authors how replay was implemented, but I cannot find their answer in the revised manuscript. I think it is possible (perhaps even likely) that the reason that SRA can improve upon "replay" is because "replay" is not implemented optimally.

- The performance of iCaRL is far below what I would expect. In my hands, on Incremental MNIST, iCaRL with $K=1000$ achieves a performance above 90%, while the authors only report ~56%. I expect that one reason for this is the way iCaRL is implemented by the authors, which for example does not use the nearest-mean-of-exemplars classification strategy. I disagree with the justification the authors give for this, as in my experience this does not hold. Again, I think it is possible (perhaps even likely) that the reason that SRA can improve upon their implementation of iCaRL is because this implementation is not optimal. (As an important side note, the authors claim in the method section that "[t]he main idea of the iCaRL algorithm is to define a memory capacity K and store K examples from previous tasks". This is obviously not true. The contribution of iCaRL lies in how to use such a memory buffer.)

- Another experimental result I have concerns about is the performance of OWM, which was included based on a suggestion from reviewer #3. As with replay and iCaRL, the reported performance of OWM is substantially lower than I would expect. For example, these two papers (<https://arxiv.org/abs/2005.03490> and <https://arxiv.org/abs/2106.08085>) both seem to find substantially stronger performance of OWM. It is possible that there are reasons for this difference (e.g., different networks architectures), but at the moment I have no way of checking that OWM was performed correctly due to incomplete method descriptions. For example, OWM has an influential hyperparameter α , but I could not find a description of how the authors set this hyperparameter.

My concerns about the seemingly low performance of these baseline methods is reinforced by the insufficient method section. Based on the provided descriptions I would not be able to replicate the key results from this paper.

Although it is certainly not a substitute, sometimes the lack of detailed methods can be alleviated by providing clear code. I tried to check the code to understand why some of the results reported in this manuscript deviate from results reported elsewhere in the literature (and/or from my own experiments). Unfortunately, although it is stated that code is available through a Google drive link and through an attached zip-file, neither the Google drive link nor the zip-file is present.

To summarise, I do not think the authors have demonstrated that SRA can improve the performance of true state-of-the-art continual learning approaches, while this claim is central to the paper.

Other remaining concerns:

- Relating to Figure 1. I welcome the changes the authors made to make the training process for this toy example more symmetric across both tasks. However, it seems to me that the training process is still not fully symmetric due to the use of momentum. If the network is trained with vanilla SGD (without momentum), does the benefit of SRA remain?

- Thanks for the addition of Supplemental Figure 3. (I would strongly recommend to replace Figure 6 by Supplemental Figure 3, as this is the more relevant comparison.) However, based on Supplemental Figure 3, it is not clear to me that, as is claimed in the text, the difference between "digit-specific" neurons and random neurons is more pronounced for the recently learned task (digits 2 and 3) than for the previously learned task (digits 0 and 1). To make this claim, the authors must perform the relevant statistical test.

Reviewer #3 (Remarks to the Author):

I appreciate the revision and detailed explanations made by the authors in addressing my previous comments, which have been largely satisfactory.

Two minor issues to be solved:

1. The title of section A "Sleep replay prevents catastrophic forgetting in ANNs" is detached from its content.
2. Line 361-363, OWM does not "need to feed all previous inputs through the network to compute projections". It stores a continually updated projector to do that.

Comments from Reviewer #2

Comment 1: *As indicated in my original review, I had doubts about the level of performance that the authors obtained using “replay”. I had asked the authors how replay was implemented, but I cannot find their answer in the revised manuscript. I think it is possible (perhaps even likely) that the reason that SRA can improve upon “replay” is because “replay” is not implemented optimally.*

Response 1: We would like to thank the reviewer for pointing out to the deficiencies of our original implementation of the “replay” methods. To address this comment we performed additional analysis of the “replay” and few other standard methods mentioned by the reviewer (specifically, methods include rehearsal, iCaRL, and OWM). We identified that the lower performance we originally reported for iCaRL was (mainly) due to the lower number of epochs used in training and also due to some differences in our implementation of iCaRL. In the modified manuscript we now report the accuracy that we believe is expected by the reviewer. The main reason for our original suboptimal implementation was that the original number of training epochs per task was set based on identifying when a network achieves optimal performance on a single task. We discovered that both OWM and “replay” methods (rehearsal, iCaRL) required more epochs per task in order to converge when multiple tasks were trained sequentially. We provide specific evidences to support this statement in Response #2 (iCaRL) and Response #3 (OWM). With that, we are now able to see the expected performance (e.g. 90% on MNIST with K = 1000 for iCaRL).

Importantly, we are glad to report that in all the cases combining SRA with “replay” methods still led to further improvements in performance (unless amount of old data was large enough so performance of “replay” methods alone approached a theoretical limit). We present these results

below in Response 2 (iCaRL) and Response 3 (OWM). Moreover, this point raises an additional benefit of incorporating SRA with a state-of-the-art rehearsal method - we can both reduce the training time and the number of images that are replayed and achieve a similar performance as a rehearsal method alone with longer training and more images.

All specific changes that we made to the text to reflect these updates are summarized at the end of our response to the Comment #2 below.

Comment 2: *The performance of iCaRL is far below what I would expect. In my hands, on Incremental MNIST, iCaRL with $K=1000$ achieves a performance above 90%, while the authors only report ~56%. I expect that one reason for this is the way iCaRL is implemented by the authors, which for example does not use the nearest-mean-of-exemplars classification strategy. I disagree with the justification the authors give for this, as in my experience this does not hold. Again, I think it is possible (perhaps even likely) that the reason that SRA can improve upon their implementation of iCaRL is because this implementation is not optimal. (As an important side note, the authors claim in the method section that “[t]he main idea of the iCaRL algorithm is to define a memory capacity K and store K examples from previous tasks”. This is obviously not true. The contribution of iCaRL lies in how to use such a memory buffer.)*

Response 2: Thank you for these comments aimed at ensuring we have implemented iCaRL optimally. As mentioned above, upon further review we have found that the difference in reported vs expected iCaRL performance was due to the lower number of epochs used in training and also due to some differences in our implementation of iCaRL, specifically due to a difference in our implementation of the nearest-mean-of-exemplars classifier. After modifying our implementation of iCaRL and observing an increase in baseline performance, we still found that SRA can be complementary to this rehearsal method and can result in equivalent classification accuracy with either: a) fewer stored images, or b) lower training time. We discuss our implementation and testing of iCaRL in more details below.

First, we would like to illustrate our new implementation of iCaRL and a few experiments we did to probe which components of iCaRL are instrumental in promoting lifelong learning. We tested iCaRL in the following conditions - random herding vs. the iCaRL herding method, distillation vs. no distillation (using hard instead of soft targets), and finally NeM classifier vs. a normal neural network classifier with softmax activation function (denoted TC). There were 8 experiments in total. We tested all these models on MNIST, CIFAR10, and Fashion MNIST datasets for memory capacities including $K = 50$, $K = 100$, $K = 200$, $K = 500$, $K = 1000$, $K = 2000$, and $K = 20000$. The results for these experiments are shown in the following figures (Figures 1-3 on MNIST, Fashion MNIST, and CIFAR10, respectively). Error bars indicate average accuracy over 5 random task orders (e.g., task order 1 = 3,4,5,1,2, task order 2 = 4,5,3,2,1). Note that in the experiments shown in Figures 1-3 the accuracy values are reported when iCaRL was trained for 10 epochs/task (see below).

Figure 1: Comparison of different methods of implementing iCaRL on MNIST for different memory capacities (K , x-axis). Each group of bars is labelled as follows: dark blue (first bar) – TC classifier with iCaRL herding and no distillation, blue (second bar) – TC classifier with random herding and no distillation, light blue (third bar) – TC + iCaRL herding and distillation, teal (fourth bar) – TC + random herding and distillation, yellow (fifth bar) – NeM classifier with iCaRL herding and no distillation, orange (sixth bar) – NeM classifier with random herding and no distillation, red (seventh bar) – NeM classifier with iCaRL herding and distillation, magenta (eighth bar) – NeM classifier with random herding and distillation.

Figure 2: Comparison of different methods of implementing iCaRL on Fashion MNIST for different memory capacities (K , x-axis). Each group of bars is labelled as follows: dark blue (first bar) – TC classifier with iCaRL herding and no distillation, blue (second bar) – TC classifier with random herding and no distillation, light blue (third bar) – TC + iCaRL herding and distillation, teal (fourth bar) – TC + random herding and distillation, yellow (fifth bar) – NeM classifier with iCaRL herding and no distillation, orange (sixth bar) – NeM classifier with random herding and no distillation, red (seventh bar) – NeM classifier with iCaRL herding and distillation, magenta (eighth bar) – NeM classifier with random herding and distillation.

no distillation, orange (sixth bar) – NeM classifier with random herding and no distillation, red (seventh bar) – NeM classifier with iCaRL herding and distillation, magenta (eighth bar) – NeM classifier with random herding and distillation.

Figure 3: Comparison of different methods of implementing iCaRL on CIFAR10 for different memory capacities (K , x-axis). Each group of bars is labelled as follows: dark blue (first bar) – TC classifier with iCaRL herding and no distillation, blue (second bar) – TC classifier with random herding and no distillation, light blue (third bar) – TC + iCaRL herding and distillation, teal (fourth bar) – TC + random herding and distillation, yellow (fifth bar) – NeM classifier with iCaRL herding and no distillation, orange (sixth bar) – NeM classifier with random herding and no distillation, red (seventh bar) – NeM classifier with iCaRL herding and distillation, magenta (eighth bar) – NeM classifier with random herding and distillation.

In all cases, the NeM classifier now performed better than the TC classifier (this was not the case in our earlier implementation of iCaRL). In terms of herding strategies, both random herding and iCaRL herding (choosing examples which minimize the difference between the exemplar set and class means) seem to perform roughly equally. Finally, we found that using hard class labels (no distillation) tends to result in slightly higher classification accuracy compared to soft class labels (distillation). However, in the following experiments we use distillation to keep our implementation of iCaRL similar to the original iCaRL implementation (Rebuffi et al, 2017). Thus, in following experiments, we now experiment with iCaRL with the exemplar herding method, NeM classifier, and distillation.

As mentioned in our Response #1, another reason for low iCaRL performance was due to the low training time used in our original paper (2-4 epochs/task). We have performed an analysis of iCaRL with different numbers of epochs/task ranging from 1 to 10 and show the result in the next figure (Figure 4). Note that each dot represents the mean of 5 network simulations (with random task order) of training each task for X Epochs/Task, where X is shown on the x-axis

(ranging from 1-10). Here, we also performed comparisons between NeM classifier (dashed lines) and TC classifier (solid lines).

As it was apparent for all 3 datasets, iCaRL requires more than 2-4 epochs per task in order to reach its optimal performance. Thus, for all new experiments (including experiments shown in Figures 1-3) we used 10 epochs per task in order to analyze the effect of SRA on an iCaRL model that is fully trained.

Figure 4: Accuracy as a function of number of epochs per task for iCaRL for 3 datasets (MNIST – top left, Fashion MNIST – top right, and CIFAR10 – bottom) and 4 different memory capacities (K=50 – red, K=100– green, K=500– black, K=1000 – blue). In addition, dashed line represents use of NeM classifier whereas solid lines indicate use of TC classifier.

Our hypothesis is that sleep can have two potential benefits when combined with the state-of-the-art rehearsal methods: 1) SRA + iCaRL can achieve the same accuracy but with less storage capacity (K) as iCaRL alone, and 2) SRA can achieve the same accuracy but with less training time as iCaRL alone. The following figure and data tables support these two claims even after we used optimal (as described above) implementation of iCaRL. In the case of MNIST, iCaRL with K = 100 achieved a performance of 65.502% after 10 epochs/task and iCaRL with K = 200 achieved a performance of 76.856% after 10 epochs/task. iCaRL+SRA with K = 100 achieved a classification performance of 78.086% after 10 epochs/task. Thus, a higher accuracy can be achieved by iCaRL + SRA when using the same or even lower (in this specific case) memory

capacity. For MNIST, Fashion MNIST and CIFAR10 datasets, in almost all cases (except $K = 2000, 5000$ for MNIST) iCaRL + SRA had higher performance than iCaRL alone when using the same memory capacity.

These results suggest that SRA can be applied to iCaRL and further improve performance without increase in storage. Complimentary to this statement, we observed that the same iCaRL base performance can be achieved with a lower memory capacity when SRA was included, suggesting that SRA can make rehearsal methods more data efficient. In general, the advantages of applying SRA were higher for lower values of K .

Table 1: Comparison of iCaRL and iCaRL + SRA on MNIST, Fashion MNIST, and CIFAR10 datasets at different memory capacities (all networks trained for 10 epochs/task)

	$K = 50$	$K = 100$	$K = 200$	$K = 500$	$K = 1000$	$K = 2000$	$K = 5000$
iCaRL, MNIST	53.428	65.502	76.856	87.326	90.628	92.850	94.698
SRA + iCaRL, MNIST	69.97	78.086	84.498	88.862	91.130	92.742	94.582
iCaRL, Fashion MNIST	49.342	57.786	62.828	69.038	73.972	78.030	80.768
SRA + iCaRL, Fashion MNIST	51.554	61.916	65.110	69.798	75.226	78.542	82.104
iCaRL, CIFAR10	35.156	43.244	49.102	54.898	59.528	62.878	66.08
SRA + iCaRL, CIFAR10	39.382	46.002	51.324	57.504	61.23	64.018	66.71

To directly support our 2nd claim that SRA can augment rehearsal methods by decreasing the training time, we present the following figure (Supplementary Figure 7 in the revised manuscript) where we plot final accuracy as a function of the number of training epochs for iCaRL alone and iCaRL+SRA. Here, we can see that in all cases, iCaRL + SRA converged more rapidly than iCaRL alone. For $K = 50$, iCaRL + SRA achieved after just one epoch the same accuracy as iCaRL alone after 10 training epochs. For $K = 100$, 4 epochs were required with iCaRL + SRA before the same 10 epoch accuracy was obtained with iCaRL alone. For $K = 1000$, iCaRL + SRA after 8 epochs had a final accuracy of 87.698%, whereas iCaRL even after 10 epochs only achieved a final accuracy of 87.32%. As reported above, in general, the benefits were higher for lower values of K . This same type of analysis was performed for Fashion

MNIST and CIFAR10 leading to the same conclusions. To quantify, we defined the training savings as a number of epochs after which iCaRL + SRA achieves a greater performance than iCaRL alone after 10 epochs. The training savings on all 3 datasets (averaged across all memory capacities and task orders) were: 3.73 epochs for MNIST, 3.67 epochs for Fashion MNIST, and 2.80 epochs for CIFAR10. In sum, these results suggest that SRA can be complementary to state-of-the-art rehearsal methods by both a) reducing the number of images, and b) reducing length of training needed to achieve similar performance.

To report all these new results, the following changes were made:

- 1) Table 1 remains unchanged - we show the performance of SRA, OWM, SRA+2% rehearsal using the same number of epochs per task as in the original paper to keep testing consistent across all the methods. However, we immediately mention that some methods' performance can be improved by using more training epochs per task and refer to the new Table 2. The following text was included:

“Note that the training time used in Table 1 was based on determining when a sequentially trained network reached its optimal performance on a single task. We, however, discovered that rehearsal methods and OWM may need longer training time in order to reach their optimal performance (see Supporting Information, Fig. 6-7 and Table 2). Thus, we hypothesize that SRA could support these rehearsal methods by reducing the training time in addition to reducing memory capacity requirements. This point is explored below.”
- 2) In the new Table 2, we show the performance of improved iCaRL implementation when the total number of epochs per task was increased to 10. We have also added updated results for OWM (see Response 3 below) to this table to show that OWM needs more epochs/task to reach an optimal performance.
- 3) We included additional details about implementation of the basic rehearsal approach in the section titled “Including old data during training” in the Methods section. Note that this method is different from iCaRL and is meant to serve a simple comparison to doing basic rehearsal during training. The following text was included:

“The exact images from old tasks were randomized and the fraction of old images stored was defined by degree of rehearsal. Thus, if task 1 has 5000 images and task 2 has 5000 images, then during training with 2% rehearsal, 2% of task 1 images

were stored (i.e., 100 random images and their corresponding hard-target labels were stored from task 1) and incorporated into the task 2 training dataset.”

- 4) We have modified the text in the main manuscript to reflect new results on iCaRL vs iCaRL+SRA performance:
“In the case of MNIST, iCaRL with $K = 100$ achieved a performance of 65.502% after 10 epochs/task and iCaRL with $K = 200$ achieved a performance of 76.856% after 10 epochs/task. iCaRL+SRA with $K = 100$ achieved a classification performance of 78.086% after 10 epochs/task. Thus, a higher accuracy can be obtained with iCaRL + SRA and it may be even achieved with a lower memory capacity. For MNIST, Fashion MNIST and CIFAR10 datasets, in almost all cases (except $K = 2000$ for MNIST) iCaRL + SRA had higher performance than iCaRL alone for the same memory capacity. We also observed that OWM could benefit from longer training times. However, the performance of iCaRL + SRA with $K = 100$ always exceeded OWM performance, suggesting that rehearsal methods are still the state-of-the-art in class-incremental continual learning settings.

Note that these experiments were run for a larger number of epochs/task as compared to Table 1. Thus, we hypothesized that in addition to lowering memory requirements, SRA could also reduce training time (denoted as number of epochs per task) needed to achieve optimal results. Indeed, we found that iCaRL + SRA converges more rapidly than iCaRL alone. For example, for $K = 50$, iCaRL + SRA achieved after just one epoch/task of training the same accuracy as iCaRL alone after 10 training epochs/task. For $K = 100$, 4 epochs/task were required with iCaRL + SRA before the same 10 epoch/task accuracy was obtained with iCaRL alone. For $K = 1000$, iCaRL + SRA after 8 epochs/task had a final accuracy of 87.698%, whereas iCaRL even after 10 epochs/task only achieved a final accuracy of 87.32%. In general, the benefits of SRA were higher for lower values of K . We defined the training savings as the number of epochs after which iCaRL + SRA achieves a greater performance than iCaRL alone after 10 epochs/task. The training savings on all 3 datasets (averaged across all memory capacities and task orders) were: 3.73 epochs/task for MNIST, 3.67 epochs/task for Fashion MNIST, and 2.80/task epochs for CIFAR10 (see Supplemental Information, Figure 7 for example plots).

In sum, we found that SRA can support various rehearsal methods by a) reducing the amount of old data that are stored (or allowing only replay of the highest-quality generated examples); b) reducing training time.”

- 5) We included Supplementary Figure 7 where we plot final accuracy as a function of the number of training epochs for iCaRL alone and iCaRL+SRA.
- 6) Our description of iCaRL was modified based on our new understanding of the method (specifically we modified the line the reviewer mentioned in his/her comment and added more detail on implementation):

“The main idea of the iCaRL algorithm is to utilize a fixed memory capacity (capacity K , where K represents number of examples stored from previous classes) in an efficient way to prevent catastrophic forgetting.”

and

“In addition to introducing an efficient exemplar management, the learning rule and classification scheme are modified in the original iCaRL implementation (Rebuffi et al., 2017). Specifically, the learning rule incorporates both soft-target distillation over examples from previous tasks as well as cross-entropy loss from the newest task. The labels for the new images are binary one-hot encoded vectors with the correct classifications. However, the labels for the old images are computed by passing these images through the network and storing the output from the network. In addition, the classification scheme is changed to use the Nearest Means of Exemplar classification strategy, where test inputs are fed through the network and their representation in the last hidden layer (before the classification layer) is used to compute the classification for that test example. Its label is determined by finding the nearest-class-mean using the exemplar set and all class labels which have been learned so far. In our implementation of iCaRL, we used the Nearest Means of Exemplar classifier as well as soft-target distillation. ”

***Comment 3:** Another experimental result I have concerns about is the performance of OWM, which was included based on a suggestion from reviewer #3. As with replay and iCaRL, the reported performance of OWM is substantially lower than I would expect. For example, these two papers (<https://arxiv.org/abs/2005.03490> and <https://arxiv.org/abs/2106.08085>) both seem to find substantially stronger performance of OWM. It is possible that there are reasons for this difference (e.g., different networks architectures), but at the moment I have no way of checking that OWM was performed correctly due to incomplete method descriptions. For example, OWM has an influential hyperparameter `alpha`, but I could not find a description of how the authors set this hyperparameter.*

Response 3: We would like to thank the reviewer again for pointing out to this discrepancy between reported here and previously published OWM performance. As mentioned above, we discovered that the duration for running OWM in the original implementation was not optimal for sequential tasks training. Indeed, although several of the networks tested in the paper converge after only 2-4 epochs per task, this was not true for OWM which, in some cases, takes longer to converge. Below (Figure 5), we show performance on the entire dataset following a class-incremental learning paradigm with 5 tasks per dataset and 2 classes per task. In the original version of the paper, we used 2 epochs per task for MNIST, Fashion MNIST, and 4 epochs per task for CIFAR10. As present in the figure below, for MNIST and Fashion MNIST, there was still room to improve OWM accuracy with more training epochs. Our final accuracy on MNIST was around 77% (up from 56%) after 10 epochs/task which aligns with the second paper mentioned by the reviewer (<https://arxiv.org/abs/2106.08085>, Figure 2, class-incremental MNIST). On Fashion MNIST, our final accuracy improves from 49% to 58%. On CIFAR10, there does not seem to be any improvement in OWM after the first epoch of training. On the Cross-Modal MNIST and CUB-200 task, the reported number of epochs in our original paper

resulted in convergence of OWM so these numbers were not updated. We found that in all cases, iCaRL + SRA with just $K = 100$ outperformed OWM.

Figure 5: Analysis of OWM performance as a function of number of epochs/task.

As for low OWM performance on CIFAR10 compared to other papers, we hypothesize a potential reason could be that our study only applies OWM to the fully connected layers, not in an end-to-end fashion, as done in the referenced papers. In these papers a feature extractor (the convolutional layers) was also trained on CIFAR10 data during incremental training, whereas we use pre-extracted CIFAR10 features and only trained the fully connected classifier. (Papers report final accuracies on CIFAR10 with 2 tasks of 52%-54%

(<https://arxiv.org/pdf/2005.03490.pdf> and <https://www.nature.com/articles/s42256-019-0080-x.pdf>).) The evidences to support our hypothesis are the following: we found that using features extracted from the same convolutional backbone as used in the paper, but trained on the CIFAR dataset (instead of Tiny ImageNet, as done in our paper), can improve OWM performance on CIFAR10 from $\sim 34\%$ to $\sim 42\%$. Thus, OWM may work well in scenarios where the feature extractor can also be fine-tuned on the dataset, whereas SRA works well in scenarios where only the fully connected classifier is modified during training. The networks tested in these other works also have more parameters ($\sim 3m$ compared to $\sim 2m$ in our network) and have higher joint training accuracies (76% in <https://arxiv.org/pdf/2005.03490.pdf>, compared to 72% in our paper).

We would also like to provide more details on our implementation of OWM. We used the GitHub repository referenced in the original OWM paper (<https://github.com/beijixiong3510/OWM>). We modified the code provided in the repository to change the network architecture and some training details (learning rate, optimizer, momentum, dropout, etc.) to be in line with our implementation of SRA and other models we compare with. On CIFAR10, we performed a layer-specific grid search over the alpha parameter. The chosen alpha values for each task can now be found in the supplement. For all tasks except CIFAR-10, we used alpha values of $0.9 \cdot (0.001^\alpha)$, 0.1^α , and 0.6 in each of the 3 hidden layers. Here, α represents the progress made through training the current task. For CIFAR-10, we used $0.9 \cdot (0.0001^\alpha)$, 0.1^α , and 0.006 as the alpha parameters.

We made the following changes to the manuscript:

- 1) OWM performance is left the same in Table 1 but a note is made that OWM could benefit from longer training times.

“Note that the training time used in Table 1 was based on determining when a sequentially trained network reached its optimal performance on a single task. We, however, discovered that rehearsal methods and OWM may need longer training time in order to reach their optimal performance (see Supporting Information, Figures 6-7 and Table 2).”

- 2) OWM performance with longer training is shown now in new Table 2 and the following description is added:

“We also observed that OWM could benefit from longer training times. However, the performance of iCaRL + SRA with $K = 100$ always exceeded OWM performance, suggesting that rehearsal methods are still the state-of-the-art in class-incremental continual learning settings.”

- 3) Figure 5 shown above is added to the supplemental information (as Figure 6 in the supporting information) as well as the following description:

“Note that in the manuscript we discuss two different versions of OWM - one with shorter training and another one with longer training. Table 1 in the manuscript shows OWM performance when trained with 2-4 epochs/task (the same as for all other methods) on MNIST, Fashion MNIST and CIFAR10. Table 2 shows OWM performance when trained with 10 epochs/task on all datasets. We discovered that OWM takes longer to converge than most of the other methods tested in this manuscript and so we present data over a range of training duration (see Figure 6). The results with longer training are in line with other implementations of OWM except for in the case of CIFAR-10 (9,10). We hypothesize that this disparity occurs because our study only applies OWM to the fully connected layers, not in an end-to-end fashion, as done in (9). Since our extracted features for CIFAR10 images were based on Tiny Imagenet, they are likely not maximally informative. If we use extracted features from a network trained on CIFAR10 instead of Tiny Imagenet, we could improve OWM performance on CIFAR10 (when trained sequentially) from 34% to 42%. This suggests that OWM may work better in scenarios where the feature extractor can also be fine-tuned on the dataset.”

- 4) Details on OWM implementation are included in Supplementary material:

“We used alpha values of $0.9 \cdot (0.001^l)$, 0.1^l , and 0.6 in each of the 3 hidden layers. Here, l represents the progress made through the training of the current task. For CIFAR-10, we used $0.9 \cdot (0.0001^l)$, 0.1^l , and 0.006 as the alpha parameters.”

Comment 4: *My concerns about the seemingly low performance of these baseline methods is reinforced by the insufficient method section. Based on the provided descriptions I would not be able to replicate the key results from this paper.*

Although it is certainly not a substitute, sometimes the lack of detailed methods can be alleviated by providing clear code. I tried to check the code to understand why some of the results reported in this manuscript deviate from results reported elsewhere in the literature (and/or from my own experiments). Unfortunately, although it is stated that code is available through a Google drive link and through an attached zip-file, neither the Google drive link nor the zip-file is present.

Response 4: We apologize for the missing Google drive link - it seems the link went missing when the document was converted to a PDF file. We double checked that the Google drive link is working now (see new link here: <https://drive.google.com/drive/folders/1LwocgKd-JaJXBmaZkoZXDxqcPpGDO4u1?usp=sharing>). We have also included our new implementation of iCaRL within this code repository. As for OWM, our code is directly sourced from the authors' Github repository (<https://github.com/beijixiong3510/OWM>). We have also modified the methods section (and supplement) in order to better describe both the implementation of SRA and these other methods (see our response to the comments above).

Comment 5: *Relating to Figure 1. I welcome the changes the authors made to make the training process for this toy example more symmetric across both tasks. However, it seems to me that the training process is still not fully symmetric due to the use of momentum. If the network is trained with vanilla SGD (without momentum), does the benefit of SRA remain?*

Response 5: Thank you for this comment. We agree that the use of momentum makes training phases to be not fully symmetric. Thus, we repeated the experiment without the use of momentum (the new figure is shown here and updated in the manuscript).

We find a few differences between the two cases (with and without momentum) but we believe that our main conclusions remain the same. In both cases, the network after SRA is able to correctly classify both tasks for a larger degree of pixel overlap, compared to the network without SRA (panel B). In Panel C, we report that after training on task 2 (TRT2), the input from T1 images to T1 output neurons (black bar) became negative (with momentum the input was positive). Since the input from T2 images to T1 output neurons (red bar) was positive, it led to incorrect classification. After SRA the input from T1 images to T1 output neurons remains negative (as we mention in the main text, since T1 neurons do not spike during sleep in this simple model, the weights to T1 output neurons do not change), however the input from T2 images to T1 output neurons became even more negative, thus leading to the correct classification. In Panel D, we still observe that the main benefit of SRA is to make input from T1 pixels to T2 output neurons more negative. The main difference, compared to the model with momentum, is that the weight range is smaller (approximately from -0.05 to 0.05), whereas before (when momentum was used) the weights could reach ~ 0.2 . We have replaced the figure in the paper with this new updated figure and clarified that there was no momentum used in the training process.

Comment 6: Thanks for the addition of Supplemental Figure 3. (I would strongly recommend to replace Figure 6 by Supplemental Figure 3, as this is the more relevant comparison.) However, based on Supplemental Figure 3, it is not clear to me that, as is claimed in the text, the difference between “digit-specific” neurons and random neurons is more pronounced for the recently learned task (digits 2 and 3) than for the previously learned task (digits 0 and 1). To make this claim, the authors must perform the relevant statistical test.

Response 6: Thank you for this suggestion. We have moved the supplemental figure showing spiking activity of digit-specific neurons when the network is presented with random inputs to the main text and removed the original figure as it provides no additional information. We have also performed the relevant statistical tests, as suggested by reviewer, comparing firing rates of digit-specific neurons with random neurons, and we found that digit-specific neurons are significantly more active than a random subset of neurons (see p-values in the figure below). In addition, we have performed statistical tests comparing firing rates of digit-specific neurons to each other. Specifically, we compared digit-0 and digit-1 (Task 1) neurons with digit-2 and digit-3 (Task 2) neurons. We found that digit-2 neurons and digit-3 neurons are significantly more active ($t(100) = 3.456$, $p = 0.004$, 1-sided t-test, Bonferroni Correction) in layer one. In layer two, this result also holds ($t(100) = 5.215$, $p < 0.001$, 1-sided t-test, Bonferroni Correction). These results have been added to the manuscript.

The following text was included:

“To compare firing rates of digit-specific neurons, we concatenated the firing rates of digit-2 and digit-3 (Task 2) neurons and compared them with the concatenated firing rates of digit-0 and digit-1 (Task 1) neurons. We found that digit-2 neurons and digit-3 neurons were significantly more active in layer one ($t(200) = 3.456$, $p = 0.004$, 1-sided t-test, Bonferroni Correction) and layer two ($t(200) = 5.215$, $p < 0.001$, 1-sided t-test, Bonferroni Correction). This suggests that spontaneous firing patterns during sleep are correlated with activity observed during task learning, in agreement with neuroscience data \cite{ji2007coordinated,rasch2013sleep}.”

Comments from Reviewer #3

Comment 1: *The title of section A “Sleep replay prevents catastrophic forgetting in ANNs” is detached from its content.*

Response 1: We agree that the title of this section does not line up well with the section’s content. We have modified the section title to: **“Sleep Replay Algorithm (SRA) implementation and testing strategy”**.

Comment 2: *Line 361-363, OWM does not “need to feed all previous inputs through the network to compute projections”. It stores a continually updated projector to do that.*

Response 2: Thank you for pointing out this error. We have removed this line from the paper.

Reviewers' comments:

Reviewer #2 (Remarks to the Author):

Thanks to the authors for another rebuttal and the updated manuscript.

During this second revision, the authors managed to catch important mistakes that substantially changed some of the initially presented results. It is great that the authors were able to find these, although it is worrying that there were such mistakes in the manuscript, especially considering that they were hidden due to insufficient method descriptions and the lack of code.

My main concern in the previous round of review was that from a practical machine learning perspective, the value of the proposed SRA was not convincingly demonstrated. I'm afraid that I still think that this is the case. In fact, the provided clarifications and added experimental details have strengthened my concerns in this regard.

In addition, the standard to which the experiments seem to have been performed is not what is expected from a Nature Communications paper. I tried to replicate two of the reported results, but in both cases I got substantially different results.

I believe SRA can in some situations alleviate catastrophic forgetting because it is able to partially correct for a bias / imbalance in the training process. I think this is interesting to some extent, but importantly I still do not think that this has value from a practical machine learning perspective, essentially because there are simpler ways to correct for or avoid such a bias / imbalance. (An example of bias-correction when storing data is not allowed is discussed here: <https://arxiv.org/abs/2008.13710>. I think it might be interesting for the authors to consider whether what their proposed SRA does is in some ways comparable to this. But this is a side note.)

For example, I think that the approaches 'rehearsal' and 'iCaRL', when implemented optimally, almost completely avoid this bias / imbalance. However, in the submitted manuscript, the approaches 'rehearsal' and 'iCaRL' are not implemented optimally. I strongly expect that the reason that SRA is shown to have a benefit, is because SRA is able to correct some of this sub-optimality. In other words, I believe the authors were able to demonstrate benefits of SRA on top of rehearsal and/or iCaRL, because these methods were not implemented optimally.

Regarding rehearsal, the authors now provided more details about how they implemented this approach. They store a small percentage of randomly selected data for each task, and then they seem to simply add this stored data to the training data of the new task, essentially creating a somewhat larger training set. This is often not an effective way of doing rehearsal as it leads to an unbalanced dataset, because there is (much) more data from the current tasks than from previous tasks. By weighting the loss on the stored data more appropriately (e.g., as a function of how many tasks have been seen so far) this imbalance can be avoided. This is critical, as I believe SRA reduces the negative effects of this imbalance. But because this imbalance can be avoided in other, simpler ways, I am not convinced the benefit of SRA is of practical value.

To back this up, I replicated the rehearsal experiment on Incremental MNIST, using the exact same settings (i.e., epochs, batch size, architecture, optimizer, learning rate, momentum, dropout) as reported by the authors, except using a more appropriate weighting of the loss on the stored data. Strikingly, while the authors reported an average test accuracy ~47% (and when using SRA this was increased to ~67%; see TABLE I), I got an average test accuracy above 90%. This suggests it is straightforward to improve upon the performance of rehearsal + SRA as reported by the authors.

Worryingly, when I then tried to replicate the rehearsal experiment reported by the authors (i.e., now using the rehearsal approach as described in the manuscript), I got an average test accuracy ~78%, instead of the ~47% reported by the authors. As indicated above, I used the exact settings as reported by the authors.

I then attempted to replicate the reported results of iCaRL on Incremental MNIST. With a memory buffer of size 50, the authors report an accuracy $\sim 53\%$. When I followed the experimental description provided by the authors, I got an accuracy $\sim 72\%$.

I'm afraid that I can only advise rejection of this paper. It is interesting that in some situations SRA seems to be able to alleviate the effects of catastrophic forgetting, but the authors have not been able to demonstrate a benefit from a practical machine learning perspective, which in my opinion would have been needed to justify publication in Nature Communications given the previous papers that the authors already published about SRA. In addition, the standard to which the reported experiments have been performed seems low: the authors found important, initially hidden mistakes in their experiments in the 2nd round of revisions, and even after correcting those both my replication attempts failed.

Comments from Reviewer #2

Comment 1:

During this second revision, the authors managed to catch important mistakes that substantially changed some of the initially presented results. In addition, the standard to which the reported experiments have been performed seems low: the authors found important, initially hidden mistakes in their experiments in the 2nd round of revisions, and even after correcting those both my replication attempts failed.

Response 1:

We would like to state that no mistakes were done or hidden in any version of our manuscript. All the discrepancies between performance numbers we report in our study and what reviewer obtained are because of differences in how methods were implemented. Specifically we identified that training duration (iCaRL, Rehearsal, and OWM), weights initialization (Rehearsal) and choice of classifier (iCaRL) have a significant impact on performance, especially when training duration was low (please see below our detailed response). We thank the reviewer for pointing out that these other methods could be implemented more efficiently to achieve higher baseline (before SRA) performance and doing so has greatly improved our paper. However, we would like to note that across all revisions, our conclusions have remained the same. In other words, the reviewer has had many critiques about our implementation of the *other* methods but the same level of criticism has not been extended to our algorithm (SRA), which is the real contribution of this study. Despite modifying our implementations of these other methods, we have never revised a conclusion about SRA's effectiveness. Any machine learning method can be implemented slightly differently and may show different performance (e.g., please look at performance results from different peer-reviewed studies of iCaRL in Table 3). However, no matter what implementation we look at across the history of this manuscript's review, SRA was shown to have

a beneficial effect. This includes this new revision where we improved implementation of rehearsal to match performance the reviewer reported and we still found significant improvements from applying SRA.

Comment 2:

I believe SRA can in some situations alleviate catastrophic forgetting because it is able to partially correct for a bias / imbalance in the training process.... To back this up, I replicated the rehearsal experiment on Incremental MNIST, using the exact same settings (i.e., epochs, batch size, architecture, optimizer, learning rate, momentum, dropout) as reported by the authors, except using a more appropriate weighting of the loss on the stored data. Strikingly, while the authors reported an average test accuracy ~47% (and when using SRA this was increased to ~67%; see TABLE I), I got an average test accuracy above 90%. This suggests it is straightforward to improve upon the performance of rehearsal + SRA as reported by the authors.

Response 2:

Please let us start saying that in the first round of reviews, we were asked to include a new figure (fig. 1) with no momentum so that training is symmetric between tasks. We did so and reported that SRA can still improve performance even if training is symmetric. Granted, this was done for a very simple “toy” model. Here we further tested effect of SRA when it was applied after training with balanced loss function for more complex models. Thus, we now tested rehearsal model with a balanced loss function and longer training (10 epochs/task instead of 2) on MNIST dataset. With that set up, and with 2% of data from old tasks stored, we obtained the same baseline performance as reviewer reported, ~89%, before applying SRA (note that with 2 epochs, we could see performance as high as 83%, see Table 2 below). Because baseline performance was already high, SRA only delivered a modest performance increase (from 89% to 92% performance) (see Table 1 below, compare row 3 and 4). However, when we reduced amount of old data we used in training, we found very significant benefits from SRA (e.g., with 0.5% of old data SRA increased performance over rehearsal alone from 64% to 89%). Importantly, this improvement is reported when a balanced loss function was used (see methods in manuscript for details). Thus, even if the training process is balanced, SRA can have a beneficial impact on network performance. Note (see also Table 2 and Response 2 below), that we found significant benefits of using balanced loss function with 2 epochs of training, but not with 10 epochs, which is what is reported in Table 1.

Table 1: SRA improves performance of rehearsal, with or without balanced loss function. Results on MNIST dataset with 2 classes per task and 5 tasks. Numbers indicate average performance on all tasks. Ten batches of training were used to ensure a complete convergence of the training before SRA was applied.

% of Old data used:	0%	0.5%	0.75%	1%	1.5%	2%
Rehearsal	19.47	62.15	77.90	80.03	84.29	86.24
Rehearsal + SRA	57.07	83.12	83.88	88.48	91.97	92.64
Rehearsal, Balanced Loss Function	19.39	64.40	76.78	78.71	85.24	88.50
Rehearsal + SRA, Balanced Loss Function	52.77	88.97	87.97	89.41	91.76	91.99

In addition, we reported that SRA provide benefit outside the context of catastrophic forgetting, e.g., when a network is undertrained (see Supplemental Figure 5) or when input is noisy and unreliable (Tadros, ICLR 2020). These points further support our statement that SRA is doing more than a symmetry correction. Biological brain can learn continuously without access to old data or using any kind of generative model to generate such data; the brain uses sleep and unsupervised replay to do it. Previous studies, including those from our lab, used biophysical brain models with synaptic plasticity to model replay during simulated sleep and reported beneficial effect of such unsupervised replay on memory performance (e.g., Wei et al., 2016, 2018, 2020; Gonzalez et al, 2021). In this new study, we proposed an implementation of the same idea (unsupervised, off-line replay) for ANNs and we found that it can significantly improve performance of the existing state-of-the-art methods.

Comment 3:

Worryingly, when I then tried to replicate the rehearsal experiment reported by the authors (i.e., now using the rehearsal approach as described in the manuscript), I got an average test accuracy ~78%, instead of the ~47% reported by the authors. As indicated above, I used the exact settings as reported by the authors.

Response 3:

We believe that reviewer did not use the exact setting we did. Unfortunately, we missed to describe our weight initialization method. To understand why reviewer obtained the performance numbers which were higher than what we reported, we performed additional analysis and we found that the main reason was the rate of convergence. To be more specific, we reported ~47% performance for rehearsal method (with 2% old data) on MNIST data, while reviewer achieved 78%. However, we used 2 training epochs and random uniform initialization of synaptic weights between -0.02 and 0.02. It turned out to be insufficient to obtain full convergence (see Table 2 below, 1st column). Using the Keras default weight initialization method and still 2 epochs of training, led to faster convergence (Table 2, 2nd column), so we obtained performance 82% for balanced and 77% for unbalanced loss function (which is similar to what reviewer reported). Increasing training duration to 10 batches brought performance to ~90% for both initialization types and both types of loss function (columns 3 and 4). Thus, we believe the main discrepancy in our results was simply the weight initialization used. The one we used could benefit from

longer training time and thus we increased the training duration for rehearsal, as was done previously for iCaRL.

Table 2: Effect of weights initialization and loss function on training time. Results on MNIST dataset with 2 classes per task and 5 tasks; rehearsal model with 2% of old data. Uniform initialization (between -0.02 and 0.02) led to slow convergence (1st column). Changing initialization strategy led to faster convergence (2d column), so the network almost reached max performance after 2 batches of training. Using 10 batches of training led to max performance that was similar regardless of initialization or type of loss function (3d and 4th columns). Numbers are reported for one specific task ordering.

	-0.02 to 0.02 (2 epochs)	Sqrt(6/(fanin + fanout)) (2 epochs)	-0.02 to 0.02 (2 epochs)	Sqrt(6/(fanin + fanout)) (10 epochs)
Balanced Loss Function	68.12	82.42	88.50	90.98
Unbalanced Loss Function	43.53	77.31	89.21	90.35

In the new manuscript we update all numbers (Table 1, see the manuscript) with longer training time and using balanced loss function for our rehearsal method. In addition, we updated Figure 3 to account for these new results. Since the weight initialization seems to be important in rate of convergence, but not in final accuracy, we kept the same weight initialization as used before. However, as we mentioned above, none of our conclusions have changed. We still find a significant positive complementary effect of incorporating SRA into the rehearsal procedure.

Comment 4:

I then attempted to replicate the reported results of iCaRL on Incremental MNIST. With a memory buffer of size 50, the authors report an accuracy ~53%. When I followed the experimental description provided by the authors, I got an accuracy ~72%.

Response 4:

We performed extensive literature search regarding iCaRL performance which we report in Table 3 below; we believe it supports the numbers we report in our paper. Specifically, while we could not find published data for K=50 on MNIST, Chaudhry et al. used K=100 and got 56% accuracy, which is significantly less than reviewer reported for K=50 (see entry #3 in the new Table 2 below). We, in fact, got 65% for K=100. De Lange et al report 88% accuracy for K=100 (see entry #8 in the Table 2), which is more than we obtained (65%), but we got higher performance numbers for K>200. Importantly, we report significant performance benefits from applying SRA on top of any iCaRL implementation we tested and for both MNIST and CIFAR10 datasets.

We should note that these studies do not necessarily use the same exact architectures as our manuscript, but we still found our results to be in line with results from the literature on most tasks and datasets. Specifically, on CIFAR10 our iCaRL our baseline performance numbers always either exceed or came close to matching results from the literature. We suspect that iCaRL is susceptible to the task order (like most catastrophic forgetting algorithms) and that some task orders (especially with smaller K) can be easier than others (note that standard deviations are larger for smaller K).

To summarize, some of our initial implementations of the baseline methods (rehearsal, iCaRL) led to slow convergence and suboptimal performance. We are grateful to reviewer for noticing that, so we improved implementations of these methods and obtained baseline performance that matches what reviewer has reported. We still show benefits of SRA when it is applied on top of a new and more optimal iCaRL or rehearsal implementation, particularly when smaller amount of old data was used. Therefore, not a single conclusion about SRA effectiveness was ever revised. Indeed, across all the versions of the paper, we show improvements from applying SRA on top of the other methods regardless of their specific implementation.

In new version of our manuscript, we have modified the entire training approach so that we can provide a fair comparison across all methods. Thus, new results on rehearsal and iCaRL and other methods are included with both a) longer training for MNIST, Fashion MNIST, and CIFAR10 and b) the balanced loss function. See new Table 1 and Figure 3 in the paper.

Table 3: Comparison of iCaRL performance reported in literature, in our study and in the review.

	Paper	Source Numbers	Our Numbers	Reviewer's Numbers
1	Buzzega, P., Boschini, M., Porrello, A., Abati, D., & Calderara, S. (2020). Dark experience for general continual learning: a strong, simple baseline. Advances in neural information processing systems , 33, 15920-15930.	Performs experiments on CIFAR10 using iCaRL with K = 200 and K = 500. Reports 49% and 47% accuracy using same task paradigm as us	49% and 55% for K = 200 and K = 500 on CIFAR10, respectively	NA
2	Shen, G., Zhang, S., Chen, X., & Deng, Z. H. (2021, July). Generative Feature Replay with Orthogonal Weight Modification for Continual Learning. In 2021 International Joint Conference on Neural Networks (IJCNN) (pp. 1-8). IEEE.	iCaRL on CIFAR10, K = 200 and K = 2000 in 5 task paradigm. Accuracies = 47% and 58%.	Accuracies = 49% and 63% for K = 200 and K = 2000 respectively	NA
3	Chaudhry, A., Dokania, P. K., Ajanthan, T., & Torr, P. H. (2018). Riemannian walk for incremental learning: Understanding forgetting and intransigence. In Proceedings of the European Conference on Computer Vision (ECCV) (pp. 532-547).	MNIST: K = 100 with 5 tasks, 55.8% final accuracy on 5 tasks. We believe it is K = 100 because they say 0.2% of data is stored.	MNIST K = 100: 65.502% final accuracy	Reviewer uses K = 50 and reports 72% accuracy.

4	Wu, Z., Baek, C., You, C., & Ma, Y. (2021). Incremental learning via rate reduction. In Proceedings of the IEEE/CVF Conference on Computer Vision and Pattern Recognition (pp. 1125-1133).	MNIST and CIFAR10 with 10 tasks, using iCaRL with K = 200. MNIST = ~75% accuracy CIFAR10 = ~50% accuracy	MNIST: 77% CIFAR10: 49% (Note, not a 1-to-1 comparison because we used 5 tasks whereas they used 10 tasks)	NA
5	Buzzega, P., Boschini, M., Porrello, A., & Calderara, S. (2021, January). Rethinking experience replay: a bag of tricks for continual learning. In 2020 25th International Conference on Pattern Recognition (ICPR) (pp. 2180-2187). IEEE.	iCaRL with K = 200, 500, 1000 on Fashion MNIST and CIFAR10 Fashion MNIST - 75%, 77%, and 78% CIFAR10 - 41%, 41%, 42%	Fashion MNIST: 62%, 69%, 73% CIFAR10: 49%, 55%, 60%	NA
6	Abati, D., Tomczak, J., Blankevoort, T., Calderara, S., Cucchiara, R., & Bejnordi, B. E. (2020). Conditional channel gated networks for task-aware continual learning. In Proceedings of the IEEE/CVF Conference on Computer Vision and Pattern Recognition (pp. 3931-3940).	MNIST with 5 tasks, K = 500, 1000, 1500, and 2000. Reports accuracies of 80-85%	We report accuracies of 87-92% for the same memory capacity range	NA
7	Munoz, D., Narváez, C., Cobos, C., Mendoza, M., & Herrera, F. (2020). Incremental learning model inspired in Rehearsal for deep convolutional networks. Knowledge-Based Systems , 208, 106460.	K = 500 and K = 5000. (Paper says 1% and 10%) MNIST: around 60% and 65%	For K = 500, we report 87% accuracy	NA
8	De Lange, M., & Tuytelaars, T. (2021). Continual prototype evolution: Learning online from non-stationary data streams. In Proceedings of the IEEE/CVF International Conference on Computer Vision (pp. 8250-8259).	MNIST and CIFAR10 (5 tasks) using iCaRL with K = 100, 200, 500, 1000, 1500, and 2000 Results presented as a graph (figure 3 in the paper) so hard to get exact value MNIST: ~82%, ~83%,	MNIST: 66%, 77%, 87%, 91% (K = 1000), 93% (K = 2000) CIFAR10: 43%, 49%, 55%, 60% (K = 1000), 63% (K = 2000)	Reviewer does K = 50 and reports 72% accuracy.

		84%, 84%, 84%, 84%		
		CIFAR10: 30%, 35%, 38%, 38%, 38%, 38%		

Summary (for Table 3):

There are 2 papers which report different baseline performance results than us, these are summarized below. Note that for all papers, our CIFAR10 performance numbers are greater or equivalent to the published numbers. For MNIST, there is 1 out of 3 sources that differs from us and only for lower values of K (which we note has a larger standard deviation and may be heavily dependent on the task order):

1. Buzzega, P., Boschini, M., Porrello, A., & Calderara, S. (2021, January). Rethinking experience replay: a bag of tricks for continual learning. In *2020 25th International Conference on Pattern Recognition (ICPR)* (pp. 2180-2187). IEEE.

We report lower numbers on Fashion MNIST but higher numbers on CIFAR10 compared with this source. Higher numbers could be explained by use of a convolutional network in this paper. In addition, towards higher K values our numbers become closer.

2. De Lange, M., & Tuytelaars, T. (2021). Continual prototype evolution: Learning online from non-stationary data streams. In *Proceedings of the IEEE/CVF International Conference on Computer Vision* (pp. 8250-8259).

For MNIST $K \leq 100$, this source reports higher numbers. For $K > 200$, we report higher numbers. For CIFAR10 we report substantially higher numbers in all cases.

REVIEWERS' COMMENTS

Reviewer #2 (Remarks to the Author):

It might be good to start by explaining why sub-optimal (or naive) implementation of “baseline methods” is an important issue. The main novel contribution of this manuscript, and the reason why I think it is potentially suitable for publication in Nature Communications, is the claim that the SRA algorithm can be used to improve the performance of SOTA continual learning techniques. The SRA algorithm itself has already been published and it was shown in previous work that it can protect memories from catastrophic forgetting (Tadros et al., 2020 ICLR; Gonzalez et al., 2020 Elife). An important claim of the current paper (and indeed a very exciting one) is that the SRA algorithm can be used to improve upon SOTA rehearsal methods. To convincingly demonstrate this, the authors should show improvements with the SRA algorithm upon SOTA implementations of rehearsal methods. It is clear that in previous submissions the implementations of rehearsal methods used by the authors were not SOTA. In addition, it seems that many of the details that were not optimally implemented, were not described in the manuscript. For example, I could not find details of the initialization of network parameters, while the authors now explain that the initialization strategy they used led to a substantially decreased performance compared to more optimal initialization strategies.

I find it surprising that the authors do not consider these clearly sub-optimal implementations mistakes.

Unfortunately, also after the latest revision, I remain skeptical about the claim that the SRA algorithm is complementary to SOTA implementations of rehearsal methods.

I can see two possible ways forward.

One option would be to remove the claim that the SRA algorithm is complementary to SOTA rehearsal methods, and instead show that the SRA algorithm is able to improve performance in certain settings (e.g., the improvements upon fine-tuning are convincing). I think this is still interesting, although it probably makes the scope of the paper less suitable for publication in Nature Communication.

Another option would be to provide more convincing evidence for the claim that the SRA algorithm can improve upon SOTA rehearsal methods. For this, my recommendation would be to use an established open source implementation (e.g., <https://github.com/aimagelab/mammoth>) and demonstrate improvements on top of that.

Reviewer #4 (Remarks to the Author):

Because I am reviewing a manuscript that has already gone through a round of revision, I will try to refrain from requesting any major structural changes.

Overall, I thought the authors did a good job showing that their algorithm offers a measurable benefit. One aspect I particularly liked about the paper is that they didn't simply show improved performance, but rather analyzed in detail why performance was improved and related the underlying mechanisms to observations from neurobiology. This aspect makes it an outstanding example of biologically inspired AI.

The replay structure seems a bit strange to me. Only pixelwise average activation is used to determine replay. This means that replay ignores all higher-order input structure (e.g., covariance between inputs). For stimuli such as natural images, most of the interesting structure is higher-order; the mean intensity in a pixel tells you almost nothing. It's impressive that such a simple replay

structure is able to be useful. But I also wonder whether this is true empirically? Does replay in the brain only depend on mean activation?

The reference to van de Ven's work is outdated; this paper (under a different title) was published in Nature Communications in 2020. More importantly, I couldn't find any substantive discussion of this work, despite its obvious relevance.

Regarding the review comments and responses from the previous round of revision: I'm confused about why reviewer 1 and the authors seem unable to get the same results. The authors should simply provide reproducible code to the reviewers, rather than try to post hoc explain why the reviewers didn't exactly reproduce what they described in the paper.

Minor:

p. 4: "for multi-modal task" -> "for the multi-modal task"

Table 1: I'm not sure why the authors are reporting standard deviation rather than standard error or a 95% confidence interval. I'm also not sure what is going in the "Inc. CUB-200" column. What are the two numbers?

All figures with error bars should have the error bars defined in the caption. Again, these should be standard error or a confidence interval, not standard deviation.

p. 8: "the most well-established in neuroscience effect of sleep on memory is augmenting memory traces for recent task" -> "the most well-established neural effect of sleep on memory is augmenting memory traces for recent tasks"

Reviewer #5 (Remarks to the Author):

I was asked by the editor to comment on the manuscript (revised draft), feedback from Reviewer 2, and the authors' response to Reviewer 2.

I congratulate both the authors, for an interesting paper, and the reviewer, for what was clearly a very rich and thorough review process.

I found the paper to be an interesting read. For my own feedback, there are some deficiencies in the coverage of relevant historical literature. The simulations could be seen as a bit "toy" in this age of video game data sets over GPU compute farms. But I think that explorations of ideas in worlds that are small enough to actually understand what is going on are useful, and that is what the authors have done. On the whole I found the article to be interesting and worthy of publication. (I hope proof reading catches numerous LaTeX open quote typos).

Reviewer 2 raised important issues and their input has significantly improved the paper. I found the authors' replies to Reviewer 2 to be adequately convincing. Given the huge numbers of assumptions and parameters involved in setting up these models, it is not surprising that the authors and reviewer see different results in some cases. This speaks to a fundamental problem with standards for replicability in the whole field, but it would be rather unfair to punish the authors of this particular article for that systemic problem (it needs to be fixed by the community as a whole). In particular I take the authors' point in rebuttal, that while some numbers have changed all conclusions have remained the same, as well made.

I suggest that the paper should be published, with perhaps a more than usually detailed acknowledgment of thanks for improvements made by the reviewer.

Point-by-point response to the reviewers' comments and concerns.

Comments from Reviewer #2

Comment #1:

It might be good to start by explaining why sub-optimal (or naive) implementation of “baseline methods” is an important issue. The main novel contribution of this manuscript, and the reason why I think it is potentially suitable for publication in Nature Communications, is the claim that the SRA algorithm can be used to improve the performance of SOTA continual learning techniques. The SRA algorithm itself has already been published and it was shown in previous work that it can protect memories from catastrophic forgetting (Tadros et al., 2020 ICLR; Gonzalez et al., 2020 Elife). An important claim of the current paper (and indeed a very exciting one) is that the SRA algorithm can be used to improve upon SOTA rehearsal methods. To convincingly demonstrate this, the authors should show improvements with the SRA algorithm upon SOTA implementations of rehearsal methods. It is clear that in previous submissions the implementations of rehearsal methods used by the authors were not SOTA. In addition, it seems that many of the details that were not optimally implemented, were not described in the manuscript. For example, I could not find details of the initialization of network parameters, while the authors now explain that the initialization strategy they used led to a substantially decreased performance compared to more optimal initialization strategies.

I find it surprising that the authors do not consider these clearly sub-optimal implementations mistakes.

Response #1:

We first would like to respond to the claim that “...it was shown in previous work that it can protect memories from catastrophic forgetting (Tadros et al., 2020 ICLR; Gonzalez et al., 2020 Elife).” In the first referred paper (Tadros et al., 2020 ICLR), we tested sleep-like replay to improve generalization to enhance ANN performance against noise and adversarial attacks. Therefore, the implementation and the goals of that study were completely different from those in our new study. In fact the old (ICLR) implementation was not even applicable to test a concept of continual learning, as we used actual training dataset, rather than noise, to drive sleep replay. Because of that, as reviewer mentioned, we also never tested the effect of combining sleep replay with other ML approaches. So, in sum, ICLR study shows or proves nothing about benefits of unsupervised sleep replay to prevent catastrophic forgetting. In the 2nd referenced paper (Gonzalez et al., 2020 Elife), we tested the idea of sleep replay to prevent memory interference in a biophysically realistic model of the thalamocortical system. The approach that was implemented there was not at all similar to the sleep algorithm presented in our new work. In fact what was published in eLife paper was not even an “algorithm” but proof of principle that spike sequences learned in awake can be replayed during sleep to minimize interference. The task tested in that work was a simple sequence completion task. While this is an important test of the concept that sleep replay can protect memories, eLife work has zero “practical” value for ML from the perspective of applying sleep-replay to ANNs. To further add to this point, there are many excellent works from computational neuroscience field that still have a very long way to be developed into algorithms that are applicable for ML and AI (e.g., the main goal of the new NSF created EFRI BRAID program is to develop such neuroscience models and concepts to ML and AI solutions).

We fully agree that our implementation of the other methods (mainly rehearsal and iCaRL) has greatly improved from one revision to the next and we thank the reviewer for his/her suggestions. In addition, it has enabled us to describe our implementations in more detail (e.g., by providing details of network initialization). Thus, the claim that “...many of the details that were not optimally implemented, were not described in the manuscript” is, we believe, no longer true as we identified factors that resulted in lower than expected baseline performance and explained these factors in the paper. Note that many of these methods, e.g., iCaRL, do not describe all implementation details, which could explain why there is such wide variability in performance numbers reported in published works (as noted in our previous response).

As for the more general point of showing that SRC (note we decided to change the name of the approach from SRA=sleep replay algorithm, to SRC=sleep replay consolidation) can improve upon performance of state-of-the-art algorithms, we never claimed that SRC overperforms iCaRL but proposed that SRC can further improve iCaRL performance and also has an advantage of not storing any old data. We greatly appreciate reviewer help to improve our baseline iCaRL implementation (which is now mentioned in the manuscript acknowledgement) and we believe we were now able to convince the reviewer that SRC can improve performance

when applied on top of SOTA iCaRL implementation. In practice, iCaRL users may not always be able to tune all the network parameters to make sure their implementation of iCaRL is SOTA. Our results show that regardless of implementation, SRC should be able to improve performance (at least in the settings tested in the paper).

Comment #2:

Unfortunately, also after the latest revision, I remain skeptical about the claim that the SRA algorithm is complementary to SOTA implementations of rehearsal methods.

Response #2:

We again thank the reviewer for his/her previous comments, providing explicit numbers to aim for in our implementations and identifying areas in which we could improve. After our previous response showing iCaRL numbers from published works, we wish there were more specific context to this comment but we do now believe that our implementation of iCaRL is optimal and SRC can deliver further improvements.

Comment #3:

I can see two possible ways forward.

One option would be to remove the claim that the SRA algorithm is complementary to SOTA rehearsal methods, and instead show that the SRA algorithm is able to improve performance in certain settings (e.g., the improvements upon fine-tuning are convincing). I think this is still interesting, although it probably makes the scope of the paper less suitable for publication in Nature Communication.

Another option would be to provide more convincing evidence for the claim that the SRA algorithm can improve upon SOTA rehearsal methods. For this, my recommendation would be to use an established open source implementation (e.g., <https://github.com/aimagelab/mammoth>) and demonstrate improvements on top of that.

Response #3:

Here, we would simply like to compare iCaRL numbers reported in our manuscript and numbers reported in a manuscript that is cited by the “mammoth” repository (please also see the table in our previous response). For the split CIFAR-10 task, with $K = 200$ or $K = 500$, we achieved baseline (before SRC) performance numbers of 49% and 55%. A manuscript cited in the repository (as a paper using the repository) achieved results of 41% and 42%, for $K = 200$ and 500, respectively (Buzzega et al., ICPR 2020). Thus our baseline iCaRL implementation has higher performance for this task than the one using the repository.

Comments from Reviewer #4

Comment #1:

The replay structure seems a bit strange to me. Only pixelwise average activation is used to determine replay. This means that replay ignores all higher-order input structure (e.g., covariance between inputs). For stimuli such as natural images, most of the interesting structure is higher-order; the mean intensity in a pixel tells you almost nothing. It's impressive that such a simple replay structure is able to be useful. But I also wonder whether this is true empirically? Does replay in the brain only depend on mean activation?

Response #1:

We would like to thank the reviewer for highlighting this important question. One of the objectives of our work is to demonstrate that replay could be driven internally and with minimal external information. Indeed in a biological brain, sleep replay is spontaneous and occurs without any external input. So, the word “replay” often used in ML rehearsal methods has a very different meaning compared to spontaneous replay in a biological brain (we discuss this topic in the recent review papers: Hayes et al., *Neural Comput.* 2021 Oct 12;33(11):2908-2950. doi: 10.1162/neco_a_01433; Kudithipudi et al., *Nature Machine Intelligence*; 2022; Vol. 4; iss. 3; pp. 196 – 210, DOI: 10.1038/s42256-022-00452-0). While the noisy input applied in SRC does not include higher-order properties, replay during SRC involves extracting such higher order interactions, which is required, as pointed out by the reviewer, to protect memories. Indeed many results from biological sleep replay suggest complex patterns and not a simple repetition of the past inputs (eg. Gupta, Anoopum S., Matthijs AA Van Der Meer, David S. Touretzky, and A. David Redish. "Hippocampal replay is not a simple function of experience." *Neuron* 65, no. 5 (2010): 695-705; also reviewed in Hayes et al.).

More specifically, the intuition of how noise-driven “unconstructed” replay allows memory consolidation is the following. Hierarchical structure of the brain, similar to multiple layers of deep neural networks, allows higher brain structures to extract progressively higher-order input features. While brain generates sleep activity spontaneously, the sleep firing patterns still reflect synaptic weight structure that was learned during training (e.g., if cell A in visual area V2 gets stronger synaptic weights from neurons B and C but a weak weight from neuron D in area V1, then during sleep, coactivation of B+C will likely activate A but coactivation of B+D or C+D will not). In other words, higher-order patterns are still extracted spontaneously during sleep. In fact, it was reported in vivo that very similar spatial patterns of population activity were observed both when the neuron fired spontaneously and when it was driven by its optimal stimulus (Tsodyks, et al., *Science*, 1999 Dec 3;286(5446):1943-6. doi: 10.1126/science.286.5446.1943).

This is what, we believe, is accomplished by the SRC algorithm. Indeed, input to the network is basically a noise with minimal (average rate) information about data (its main goal is to induce network activation), however, all the needed information is already present in the structure of

synaptic weights and higher-order patterns are extracted spontaneously during reactivation. This, we believe, is a key reason why the brain can consolidate new and reconsolidate old memories without access to the past data.

We included a short statement to discuss this issue:

Why a noise driven reactivation, as proposed in the SRC here, can recover memories, including higher order associations learned during training? In biology, while the brain generates sleep activity spontaneously, the sleep firing patterns still reflect synaptic weight structure that was learned during training. In fact, it was reported in vivo that similar spatial patterns of population activity were observed both when the neuron fired spontaneously and when it was driven by its optimal stimulus (Tsodyks, et al., Science, 1999 Dec 3;286(5446):1943-6. doi: 10.1126/science.286.5446.1943). While the noisy input applied in the SRA may not include higher-order structure found in the training data, replay during sleep involves extracting such higher-order interactions, because the information is already present in the patterns of synaptic weights. Many results from biological sleep replay suggest complex patterns and not a simple repetition of the past inputs (Gupta, Anoopum S., Matthijs AA Van Der Meer, David S. Touretzky, and A. David Redish. "Hippocampal replay is not a simple function of experience." *Neuron* 65, no. 5 (2010): 695-705). From that perspective “spontaneous replay” found in the biological brain and implemented in the SRC here, is very different compared to “explicit replay” of the past inputs often found in ML rehearsal methods (reviewed in (Hayes et al., *Neural Comput.* 2021 Oct 12;33(11):2908-2950; Kudithipudi et al., *Nature Machine Intelligence*; 2022; Vol. 4; iss. 3; pp. 196 – 210)).

Comment #2:

The reference to van de Ven's work is outdated; this paper (under a different title) was published in Nature Communications in 2020. More importantly, I couldn't find any substantive discussion of this work, despite its obvious relevance.

Response #2:

Thank you for pointing this out. We removed the Arxiv reference from our bibliography and ensured that all in-text citations point to the Nature Communications reference. We discuss this work in the context of other “generative” approaches to continual learning as there are several such works (e.g., see also “Fearnnet: Brain-inspired model for incremental learning”). Thus, we do not discuss this work alone in detail, since several other studies utilize a similar approach (although van de Ven’s manuscript does importantly improve upon other works by combining several different approaches like context-gating and internal replay). The main point of our discussion was that although generative replay algorithms currently perform near the top of continual learning algorithms, they do not closely mimic principles used by the brain (e.g.,

back-propagation vs. local Hebbian plasticity) and thus there is still value in implementing a Hebbian-based learning rules and different replays concepts.

Comment #3:

Regarding the review comments and responses from the previous round of revision: I'm confused about why reviewer 1 and the authors seem unable to get the same results. The authors should simply provide reproducible code to the reviewers, rather than try to post hoc explain why the reviewers didn't exactly reproduce what they described in the paper.

Response #3:

We thank you for this comment. We did provide code to the reviewer and since the reviewer pointed out differences in our implementation, we tried to figure out the reasons for these discrepancies. We are not sure if the reviewer used our code but, in any case, we will be providing a link to a Github repository in the final version.

Minor Comments:

p. 4: "for multi-modal task" -> "for the multi-modal task"

Table 1: I'm not sure why the authors are reporting standard deviation rather than standard error or a 95% confidence interval. I'm also not sure what is going in the "Inc. CUB-200" column. What are the two numbers?

All figures with error bars should have the error bars defined in the caption. Again, these should be standard error or a confidence interval, not standard deviation.

p. 8: "the most well-established in neuroscience effect of sleep on memory is augmenting memory traces for recent task" -> "the most well-established neural effect of sleep on memory is augmenting memory traces for recent tasks"

Response to Minor Comments:

Thank you for these comments - we have modified the sentence structure and modified error bars in the figures to represent SEM rather than SD. As for the tables, since we do not perform direct statistical comparisons, we leave it as standard deviation as this provides more information about how different task orders can result in variance in final performance. Either way, we provide enough information to move back and forth between confidence interval, SEM, and SD. In addition, we now explicitly describe that the two numbers in the CUB-200 table represent performance on Task 1 and Task 2 (e.g., performance is separated by task to see how performance varies across the different tasks). Note that both tasks are the same size so the overall accuracy can be calculated by averaging the two values.

Comments from Reviewer #4

Comment #1:

I found the paper to be an interesting read. For my own feedback, there are some deficiencies in the coverage of relevant historical literature. The simulations could be seen as a bit "toy" in this age of video game data sets over GPU compute farms. But I think that explorations of ideas in worlds that are small enough to actually understand what is going on are useful, and that is what the authors have done. On the whole I found the article to be interesting and worthy of publication. (I hope proof reading catches numerous LaTeX open quote typos).

Response #1:

We are very happy the reviewer found the paper to be interesting and worthy of publication. We have fixed the open quote typos and added literature related to the historical coverage of continual learning in the introduction and discussion. We agree that our manuscript is not necessarily aimed at delivering a broad coverage of different tasks and algorithms but more so at exploring the interplay between neuroscience and continual learning in AI - hence the decision to focus on understandable settings. We also provide a broad overview on the rehearsal vs regularization methods in the Discussion.

Comment #2:

I suggest that the paper should be published, with perhaps a more than usually detailed acknowledgment of thanks for improvements made by the reviewer.

Response #2:

We have included a thanks in the acknowledgements for the many rounds of review that greatly improved our work.